# Automated long-term recording and analysis of neural activity in behaving animals

Ashesh K Dhawale[1,2†]*, Rajesh Poddar[1,2†], Steffen BE Wolff[1,2], Valentin A Normand[1,2], Evi Kopelowitz[1,2], Bence P Ölveczky[1,2]*

[1]Department of Organismic and Evolutionary Biology, Harvard University, Cambridge, United States; [2]Center for Brain Science, Harvard University, Cambridge, United States

**Abstract** Addressing how neural circuits underlie behavior is routinely done by measuring electrical activity from single neurons in experimental sessions. While such recordings yield snapshots of neural dynamics during specified tasks, they are ill-suited for tracking single-unit activity over longer timescales relevant for most developmental and learning processes, or for capturing neural dynamics across different behavioral states. Here we describe an automated platform for continuous long-term recordings of neural activity and behavior in freely moving rodents. An unsupervised algorithm identifies and tracks the activity of single units over weeks of recording, dramatically simplifying the analysis of large datasets. Months-long recordings from motor cortex and striatum made and analyzed with our system revealed remarkable stability in basic neuronal properties, such as firing rates and inter-spike interval distributions. Interneuronal correlations and the representation of different movements and behaviors were similarly stable. This establishes the feasibility of high-throughput long-term extracellular recordings in behaving animals.

DOI: https://doi.org/10.7554/eLife.27702.001

**\*For correspondence:**
ashesh.dhawale@gmail.com
(AKD);
olveczky@fas.harvard.edu (BPÖ)

†These authors contributed equally to this work

**Competing interests:** The authors declare that no competing interests exist.

## Introduction

The goal of systems neuroscience is to understand how neural activity generates behavior. A common approach is to record from neuronal populations in targeted brain areas during experimental sessions while subjects perform designated tasks. Such intermittent recordings provide brief 'snapshots' of task-related neural dynamics (*Georgopoulos et al., 1986*; *Hanks et al., 2015*; *Murakami et al., 2014*), but fail to address how neural activity is modulated outside of task context and across a wide range of active and inactive behavioral states, (for exceptions see [*Ambrose et al., 2016*; *Evarts, 1964*; *Gulati et al., 2014*; *Hengen et al., 2016*; *Hirase et al., 2001*; *Lin et al., 2006*; *Mizuseki and Buzsáki, 2013*; *Santhanam et al., 2007*; *Wilson and McNaughton, 1994*]). Furthermore, intermittent recordings are ill-suited for reliably tracking the same neurons over time (*Dickey et al., 2009*; *Emondi et al., 2004*; *Fraser and Schwartz, 2012*; *McMahon et al., 2014a*; *Santhanam et al., 2007*; *Tolias et al., 2007*), making it difficult to discern how neural activity and task representations are shaped by developmental and learning processes that evolve over longer timescales (*Ganguly et al., 2011*; *Jog et al., 1999*; *Lütcke et al., 2013*; *Marder and Goaillard, 2006*; *Peters et al., 2014*; *Singer et al., 2013*).

Addressing such fundamental questions would be greatly helped by recording neural activity and behavior continuously over days and weeks in freely moving animals. Such longitudinal recordings would allow the activity of single neurons to be followed over more trials, experimental conditions, and behavioral states, thus increasing the power with which inferences about neural function can be

made (*Lütcke et al., 2013*; *McMahon et al., 2014a*). Recording continuously outside of task context would also reveal how task-related neural dynamics and behavior are affected by long-term performance history (*Bair et al., 2001*; *Chaisanguanthum et al., 2014*; *Morcos and Harvey, 2016*), changes in internal state (*Arieli et al., 1996*), spontaneous expression of innate behaviors (*Aldridge and Berridge, 1998*), and replay of task-related activity in different behavioral contexts (*Carr et al., 2011*; *Foster and Wilson, 2006*; *Gulati et al., 2014*; *Wilson and McNaughton, 1994*).

While in vivo calcium imaging allows the same neuronal population to be recorded intermittently over long durations (*Huber et al., 2012*; *Liberti et al., 2016*; *Peters et al., 2014*; *Rose et al., 2016*; *Ziv et al., 2013*), photobleaching, phototoxicity, and cytotoxicity (*Grienberger and Konnerth, 2012*; *Looger and Griesbeck, 2012*), as well as the requirements for head-restraint in many versions of such experiments (*Dombeck et al., 2007*; *Huber et al., 2012*; *Peters et al., 2014*), make the method unsuitable for continuous long-term recordings. Calcium imaging also has relatively poor temporal resolution (*Grienberger and Konnerth, 2012*; *Vogelstein et al., 2009*), limiting its ability to resolve precise spike patterns (*Vogelstein et al., 2010*; *Yaksi and Friedrich, 2006*) (but see *Gong et al., 2015* for an alternative high-speed voltage sensor). In contrast, extracellular recordings using electrode arrays can measure the activity of many single neurons simultaneously with sub-millisecond resolution (*Buzsáki, 2004*). Despite the unique benefits of continuous (24/7) long-term electrical recordings, they are not routinely performed. A major reason is the inherently laborious and difficult process of reliably and efficiently tracking the activity of single units from such longitudinal datasets (*Einevoll et al., 2012*), wherein firing rates of individual neurons can vary over many orders of magnitude (*Hromádka et al., 2008*; *Mizuseki and Buzsáki, 2013*) and spike waveforms change over time (*Dickey et al., 2009*; *Emondi et al., 2004*; *Fraser and Schwartz, 2012*; *Okun et al., 2016*; *Santhanam et al., 2007*; *Tolias et al., 2007*).

To address this, we designed and deployed a modular and low-cost recording system that enables fully automated long-term continuous extracellular recordings from large numbers of neurons in freely behaving rodents engaged in natural behaviors and prescribed tasks. To efficiently parse the large streams of neural data, we developed an unsupervised spike-sorting algorithm that automatically processes the raw data from electrode array recordings, and clusters spiking events into putative single units, tracking their activity over long timescales. We validated our algorithm on ground-truth datasets and found that it surpassed the performance of spike-sorting methods that assume stationarity of spike waveforms. We used this integrated system to record from motor cortex and striatum continuously over several months, and address an ongoing debate (*Clopath et al., 2017*; *Lütcke et al., 2013*) about whether the brain is stable (*Ganguly and Carmena, 2009*; *Greenberg and Wilson, 2004*; *McMahon et al., 2014b*; *Peters et al., 2014*; *Rose et al., 2016*) or not (*Carmena et al., 2005*; *Huber et al., 2012*; *Liberti et al., 2016*; *Mankin et al., 2012*; *Morcos and Harvey, 2016*; *Rokni et al., 2007*; *Ziv et al., 2013*) over long timescales, an issue that has been previously addressed using intermittent calcium imaging (*Huber et al., 2012*; *Liberti et al., 2016*; *Morcos and Harvey, 2016*; *Peters et al., 2014*; *Rose et al., 2016*; *Ziv et al., 2013*) and extracellular recordings (*Carmena et al., 2005*; *Ganguly and Carmena, 2009*; *Greenberg and Wilson, 2004*; *Mankin et al., 2012*; *McMahon et al., 2014b*; *Rokni et al., 2007*). Our continuous, long-term recordings revealed a remarkable stability in basic neuronal properties of isolated single units, such as firing rates and inter-spike interval distributions. Interneuronal correlations and movement tuning across a range of behaviors were similarly stable over several weeks.

## Results

### Infrastructure for automated long-term neural recordings in behaving animals

We developed experimental infrastructure for continuous long-term extracellular recordings in behaving rodents (*Figure 1A*, *Figure 1—figure supplement 1*). Our starting point was ARTS, a fully Automated Rodent Training System we previously developed (*Poddar et al., 2013*). In ARTS, the animal's home-cage doubles as the experimental chamber, making it a suitable platform for continuous long-term recordings.

To ensure that animals remain reliably and comfortably connected to the recording apparatus over months-long experiments, we designed a variation on the standard tethering system that allows

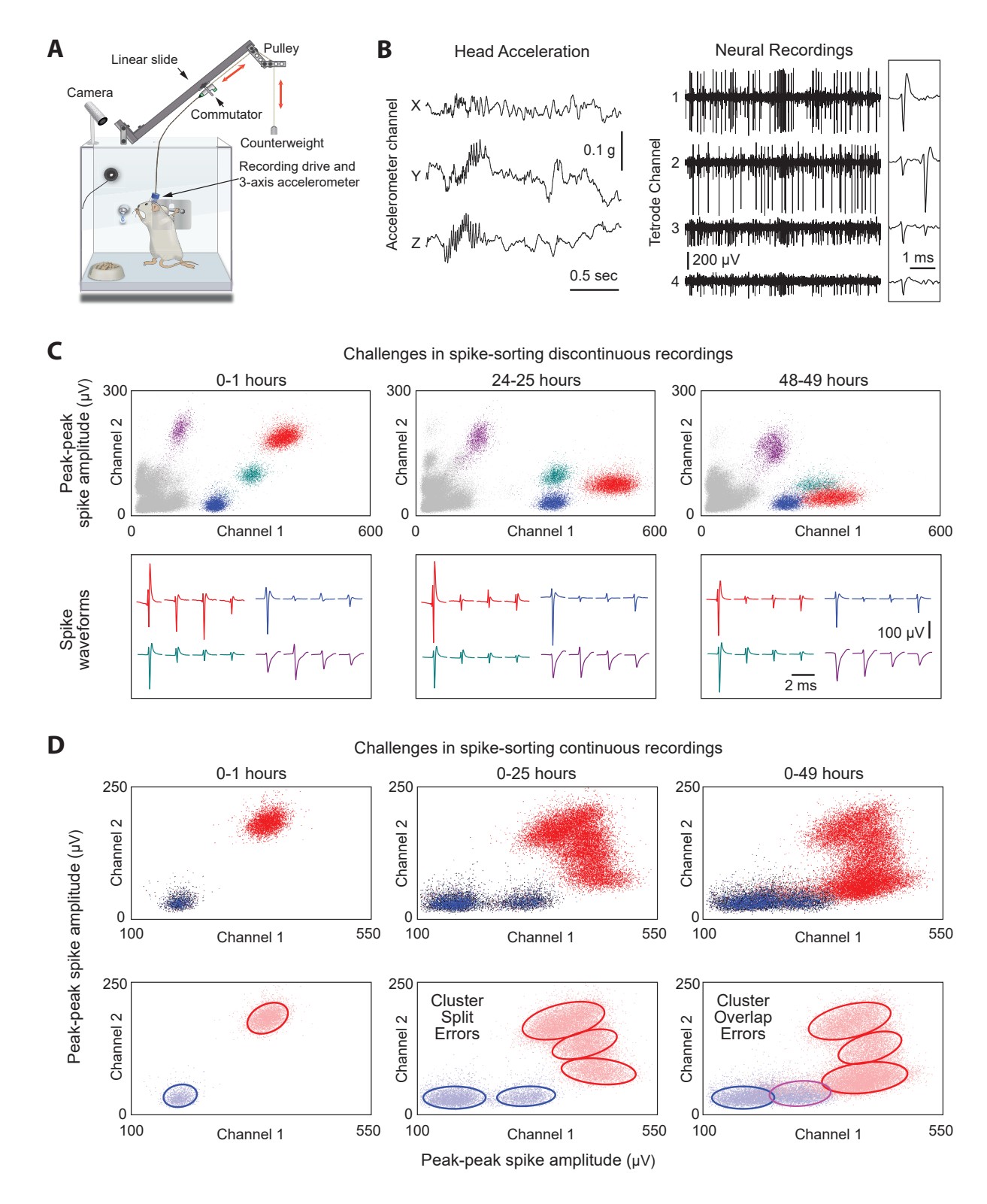

**Figure 1.** Long-term continuous neural and behavioral recordings in behaving rodents pose challenges for traditional methods of spike-sorting. (**A**) Adapting our automated rodent training system (ARTS) for long-term electrophysiology. Rats engage in natural behaviors and prescribed motor tasks in their home-cages, while neural data is continuously acquired from implanted electrodes. The tethering cable connects the head-stage to a commutator mounted on a carriage that moves along a low-friction linear slide. The commutator-carriage is counterweighted to eliminate slack in the

*Figure 1 continued on next page*

*Figure 1 continued*

tethering cable. Behavior is continuously monitored and recorded using a camera and a 3-axis accelerometer. (**B**) Example of a recording segment showing high-resolution behavioral and neural data simultaneously acquired from a head-mounted 3-axis accelerometer (left) and a tetrode (right) implanted in the motor cortex, respectively. (Inset) A 2 ms zoom-in of the tetrode recording segment. (**C**) Drift in spike waveforms over time make it difficult to identify the same units across discontinuous recording sessions. Peak-to-peak spike amplitudes (top) and spike waveforms (bottom) for four distinct units on the same tetrode for hour-long excerpts, at 24 hr intervals, from a representative long-term continuous recording in the rat motor cortex 4 months after electrode implantation. Different units are indicated by distinct colors. We tracked units over days using a novel spike-sorting algorithm we developed to cluster continuously recorded neural data (see *Figure 2*). (**D**) Continuous extracellular recordings pose challenges for spike-sorting methods assuming stationarity in spike shapes. (Top) Peak-to-peak spike amplitudes of two continuously recorded units (same as in C) accumulated over 1 hr (left), 25 hr (middle) and 49 hr (right). (Bottom) Drift in spike waveforms can lead to inappropriate splitting (middle, right panels) of single-units and/or merging (right panel) of distinct units, even though these two units are separable in the hour-long 'sessions' shown in C.

DOI: https://doi.org/10.7554/eLife.27702.002

The following figure supplement is available for figure 1:

**Figure supplement 1.** Overview of custom-built hardware for extracellular recordings in behaving rodents.

DOI: https://doi.org/10.7554/eLife.27702.003

experimental subjects to move around freely while preventing them from reaching for (and chewing) the signal cable (*Figure 1A*; see Materials and methods for details). Our solution connects the implanted recording device via a cable to a passive commutator (*Sutton and Miller, 1963*) attached to a carriage that rides on a low-friction linear slide. The carriage is counterweighted by a pulley, resulting in a small constant upwards force (<10 g) on the cable that keeps it taut and out of the animal's reach without unduly affecting its movements. The recording extension can easily be added to our custom home-cages, allowing animals that have been trained, prescreened, and deemed suitable for recordings to be implanted with electrode drives, and placed back into their familiar training environment (i.e. home-cage) for recordings.

Extracellular signals recorded in behaving animals from 16 implanted tetrodes at 30 kHz sampling rate are amplified and digitized on a custom-designed head-stage (*Figure 1B* and *Figure 1—figure supplement 1*, Materials and methods). To characterize the behavior of animals throughout the recordings, the head-stage features a 3-axis accelerometer that measures head movements at high temporal resolution (*Venkatraman et al., 2010*) (*Figure 1A–B*). We also record continuous video of the rats' behavior with a wide-angle camera (30 fps) above the cage (Materials and methods). The large volumes of behavioral and neural data (~0.5 TB/day/rat) are streamed to custom-built high-capacity servers (*Figure 1—figure supplement 1*).

## A fast automated spike tracker (FAST) for long-term neural recordings

Extracting single-unit spiking activity from raw data collected over weeks and months of continuous extracellular recordings presents a significant challenge for which there is currently no adequate solution. Parsing such large datasets must necessarily rely on automated spike-sorting methods. These face three major challenges (*Rey et al., 2015a*). First, they must reliably capture the activity of simultaneously recorded neurons whose firing rates can vary over many orders of magnitude (*Hromádka et al., 2008*; *Mizuseki and Buzsáki, 2013*). Second, they have to contend with spike shapes from recorded units changing significantly over time (*Figure 1C*) (*Dickey et al., 2009*; *Emondi et al., 2004*; *Fraser and Schwartz, 2012*). This can lead to sorting errors such as incorrect splitting of a single unit's spikes into multiple clusters, or incorrect merging of spikes from multiple units in the same cluster (*Figure 1D*). Third, automated methods must be capable of processing very large numbers of spikes in a reliable and efficient manner (in our experience, >$10^{10}$ spikes per rat over a time span of 3 months for 64 channel recordings).

Here we present an unsupervised spike-sorting algorithm that meets these challenges and tracks single units over months-long timescales. Our Fast Automated Spike Tracker (FAST) (*Poddar et al., 2017*) comprises two main steps (*Figure 2*). First, to compress the datasets and normalize for large variations in firing rates between units, it applies 'local clustering' to create de-noised representations of spiking events in the data (*Figure 2A–C*). In a second step, FAST chains together de-noised spikes belonging to the same putative single unit over time using an integer linear programming algorithm (*Figure 2D*) (*Vazquez-Reina et al., 2011*). FAST is an efficient and modular data processing pipeline that, depending on overall spike rates, runs two to three times faster on our custom

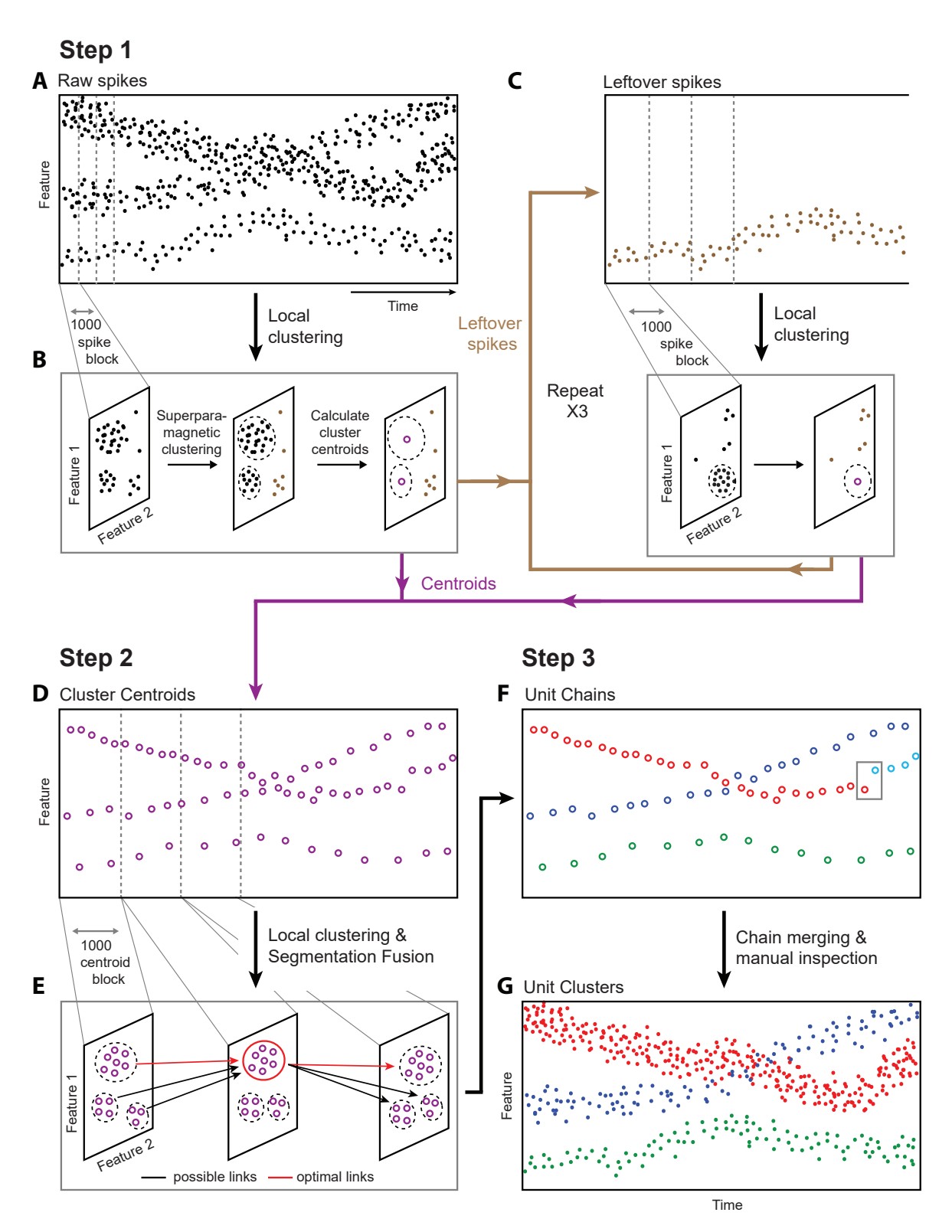

**Figure 2.** Overview of fast automated spike tracker (FAST), an unsupervised algorithm for spike-sorting continuous, long-term recordings. (**A–C**) Step 1 of FAST. (**A**) Cartoon plot of spike waveform feature (such as amplitude) over time. Spikes are grouped into consecutive blocks of 1000 (indicated by gray dashed lines). (**B**) Superparamagnetic clustering is performed on each 1000 spike-block to compress and de-noise the raw spike dataset. Centroids (indicated by purple circles) of high-density clusters comprising more than 15 spikes are calculated. These correspond to units with higher firing rates.

*Figure 2 continued on next page*

*Figure 2 continued*

(C) Leftover spikes in low-density clusters (indicated by brown dots), corresponding to units with lower firing rates, are pooled into 1000 spike blocks and subject to the local clustering step. This process is repeated for a total of 4 iterations in order to account for spikes from units across a wide range of firing rates. Cluster centroids representing averaged spike waveforms from each round of local clustering are carried forward to Step 2 of FAST. (D–E) Step 2 of FAST. (D) Centroids from all rounds of local clustering in Step 1 are pooled into blocks of size 1000 (dashed grey lines) and local superparamagnetic clustering is performed on each block. (E) The resulting clusters are linked across time using a segmentation fusion algorithm to yield cluster-chains corresponding to single units. (F) Step 3 of FAST. In the final step, the output of the automated sorting (top) is visually inspected and similar chains merged across time to yield the final single-unit clusters (bottom).

DOI: https://doi.org/10.7554/eLife.27702.004

The following figure supplements are available for figure 2:

**Figure supplement 1.** Algorithm for identifying spikes from tetrode recordings.

DOI: https://doi.org/10.7554/eLife.27702.005

**Figure supplement 2.** Algorithms for local clustering of spike waveforms and linking cluster trees to track units over time (Steps 1 and 2 of FAST).

DOI: https://doi.org/10.7554/eLife.27702.006

**Figure supplement 3.** Recommended workflow for manual verification step of FAST.

DOI: https://doi.org/10.7554/eLife.27702.007

**Figure supplement 4.** Effect of median subtraction on recording noise in behaving rats.

DOI: https://doi.org/10.7554/eLife.27702.008

built storage servers (*Figure 1—figure supplement 1*) than the rate at which the data (64 electrodes) is acquired. This implies that FAST could, in principle, also be used for online sorting, although we are currently running it offline.

To parse and compress the raw data, FAST first identifies and extracts spike events ('snippets') by bandpass filtering and thresholding each electrode channel (Materials and methods, *Figure 2—figure supplement 1*). Four or more rounds of clustering are then performed on blocks of 1000 consecutive spike 'snippets' by means of an automated superparamagnetic clustering routine, a step we call 'local clustering' (*Blatt et al., 1996*; *Quiroga et al., 2004*) (Materials and methods, *Figure 2B* and *Figure 2—figure supplement 2A–D*). Spikes in a block that belong to the same cluster are replaced by their centroid, a step that effectively de-noises and compresses the data by representing groups of similar spikes with a single waveform. The number of spikes per block was empirically determined to balance the trade-off between computation time and accuracy of superparamagnetic clustering (see Materials and methods). The goal of this step is not to reliably find all spike waveforms associated with a single unit, but to be reasonably certain that the waveforms being averaged over are similar enough to be from the same single unit.

Due to large differences in firing rates between units, the initial blocks of 1000 spikes will be dominated by high firing rate units. Spikes from more sparsely firing cells that do not contribute at least 15 spikes to a cluster in a given block are carried forward to the next round of local clustering, where previously assigned spikes have been removed (Materials and methods, *Figure 2C*, *Figure 2—figure supplement 2A–D*). Applying this method of pooling and local clustering sequentially four times generates a de-noised dataset that accounts for large differences in the firing rates of simultaneously recorded units (*Figure 2C*, Materials and methods).

The second step of the FAST algorithm is inspired by an automated method ('segmentation fusion') that links similar elements over cross-sections of longitudinal datasets in a globally optimal manner (Materials and methods, *Figure 2D–E*, *Figure 2—figure supplement 2E–F*). Segmentation fusion has been used to trace processes of individual neurons across stacks of two-dimensional serial electron-microscope images (*Kasthuri et al., 2015*; *Vazquez-Reina et al., 2011*). We adapted this method to link similar de-noised spikes across consecutive blocks into 'chains' containing the spikes of putative single units over longer timescales (*Figure 2F*). This algorithm allows us to automatically track the same units over days and weeks of recording.

In a final post-processing and verification step, we use a semi-automated method (*Dhawale et al., 2017a*) to link 'chains' belonging to the same putative single unit together, and perform visual inspection of each unit (*Figure 2F–G* and *Figure 2—figure supplement 3*). A detailed description of the various steps involved in the automated spike-sorting can be found in Materials and methods. Below, we describe how we validated the spike-sorting performance of FAST using ground-truth datasets.

## Validation of FAST on ground-truth datasets

To measure the spike-sorting capabilities of FAST, we used a publicly available dataset (*Harris et al., 2000*; *Henze et al., 2009*) comprising paired intracellular and extracellular (tetrode) recordings in the anesthetized rat hippocampus. The intracellular recordings can be used to determine the true spike-times for any unit also captured simultaneously on the tetrodes (*Figure 3A*). To benchmark the performance of FAST, we compared its error rate to that of the best ellipsoidal error rate (BEER), a measure which represents the optimal performance achievable by standard clustering methods (*Harris et al., 2000*) (see Materials and methods). We found that FAST performed as well as BEER on these relatively short (4–6 min long) recordings (*Figure 3B* and *Figure 3—figure supplement 1A*, n = 5 recordings).

Next we wanted to quantify FAST's ability to isolate and track units over time despite non-stationarity in their spike waveforms. Due to the unavailability of experimental ground-truth datasets recorded continuously over days to weeks-long timescales, we constructed synthetic tetrode recordings (*Dhawale et al., 2017b*; *Martinez et al., 2009*) from eight distinct units whose spike amplitudes fluctuated over 256 hr (10.7 days) following a geometric random walk process (*Rossant et al., 2016*) (*Figure 3C*, *Figure 3—figure supplement 2*, and Materials and methods). To make these synthetic recordings as realistic as possible, we measured several parameters including the variance of the random walk process, distribution of spike amplitudes, and degree of signal-dependent noise using our own long-term recordings in the rodent motor cortex and striatum (see *Figure 3—figure supplement 2*, Materials and methods, and the section 'Continuous long-term recordings from striatum and motor cortex'). We then compared the sorting performance of FAST to BEER on increasingly larger subsets – from 1 to the full 256 hr – of the synthetic recordings (n = 48 units from 6 simulated tetrodes). Our expectation was for the performance of BEER to, on average, decrease with accumulated time in the recording as drift increases the likelihood of overlap between different units in spike waveform feature space, and that is also what we observed (*Figure 3D*). In contrast, the performance of FAST, which was comparable to BEER for short recording durations, significantly outperformed it when the recordings spanned days (*Figure 3D*). FAST was also found to perform better than BEER over recording durations longer than 64 hr for single electrode recording configurations that are commonly used in non-human primates (*Figure 3—figure supplement 1B*). The large differences in the sorting error rates between the experimental (*Figure 3B*) and synthetic datasets (*Figure 3D*) can be largely attributed to relatively small spike-amplitudes (<100 μV) (*Harris et al., 2000*) and, consequently, lower signal-to-noise ratio (SNR = 8.1 ± 5.3, mean ±SD, n = 5 units) of units identified in both intracellular and extracellular recordings in comparison to our in vivo extracellular recordings on which the simulated datasets are modeled (SNR = 15.0 ± 7.3, n = 2572 units, *Figure 4F*).

Having validated FAST's spike-sorting performance on both experimental and synthetic datasets, we next describe how our algorithm parses data acquired from continuous long-term tetrode recordings, but we note that it can be adapted to efficiently analyze other types of electrode array or single electrode recordings – whether continuous or intermittent.

## Continuous long-term recordings from striatum and motor cortex

To demonstrate the utility of our experimental platform and analysis pipeline for long-term neural recordings, we implanted tetrode drives (16 tetrodes, 64 channels) into dorsolateral striatum (n = 2) or motor cortex (n = 3) of Long Evans rats (Materials and methods). We recorded electrophysiological and behavioral data continuously (*Figure 4A*), with only brief interruptions, for more than 3 months. We note that our recordings terminated not because of technical issues with the implant or recording apparatus, but because we deemed the experiments to have reached their end points.

We used our automated spike-sorting method (FAST) to cluster spike waveforms into putative single units, isolating a total of 1550 units from motor cortex and 1477 units from striatum (*Figure 4*). On average, we could track single units over several days (mean: 3.2 days for motor cortex and 7.4 days for striatum), with significant fractions of units tracked continuously for more than a week (11% and 36% in motor cortex and striatum, respectively) and even a month (0.4% in motor cortex and 1.7% in striatum, *Figure 4B*). Periods of stable recordings were interrupted by either intentional advancement of the electrode drive or spontaneous 'events' likely related to the sudden movement of the electrodes (*Figure 4C*). On average, we recorded simultaneously from 19 ± 15 units in motor

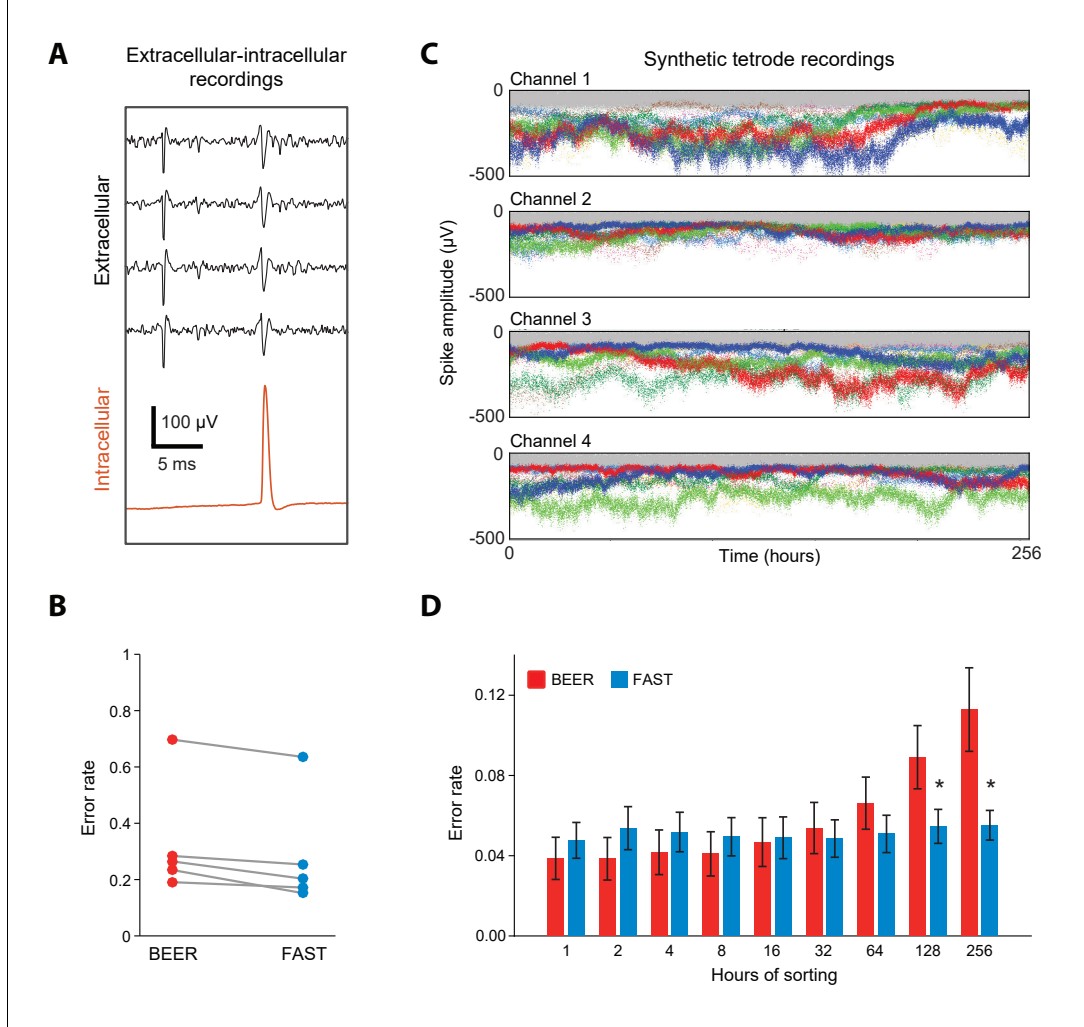

**Figure 3.** Validation of spike-sorting performance of FAST using ground-truth datasets. (**A**) Example traces of simultaneous extracellular tetrode (black) and intracellular electrode (red) recordings from the anesthetized rat hippocampus (*Harris et al., 2000*; *Henze et al., 2009*). The intracellular trace can be used to identify ground-truth spike times of a single unit recorded on the tetrode. (**B**) Spike-sorting error rate of FAST on the paired extracellular-intracellular recording datasets (n = 5) from (*Henze et al., 2009*), in comparison to the best ellipsoidal error rate (BEER), a measure of the optimal performance of standard spike-sorting algorithms (see Materials and methods, *Harris et al., 2000*). The error rate was calculated by dividing the number of misclassified spikes (sum of false positives and false negatives) by the number of ground-truth spikes for each unit. (**C**) A representative synthetic tetrode recording dataset in which we model realistic fluctuations in spike amplitudes over 256 hr (see Materials and methods). The four plots show simulated spike amplitudes on the four channels of a tetrode. Colored dots indicate spikes from eight distinct units while gray dots represent multi-unit background activity. For visual clarity, we have plotted the amplitude of every 100[th] spike in the dataset. (**D**) Spike-sorting error rates of FAST (blue bars) applied to the synthetic tetrode datasets (n = 48 units from six tetrodes), in comparison to the BEER measure (red bars) over different durations of the simulated recordings. Error-bars represent standard error of the mean. * indicates p<0.05 after applying the Šidák correction for multiple comparisons.

DOI: https://doi.org/10.7554/eLife.27702.009

The following figure supplements are available for figure 3:

**Figure supplement 1.** Validation of spike-sorting performance of FAST using ground-truth extracellular-intracellular recordings and synthetic single electrode datasets.
DOI: https://doi.org/10.7554/eLife.27702.010
**Figure supplement 2.** Analyzing long-term recordings in the DLS and MC to characterize variation in unit spike amplitude.
DOI: https://doi.org/10.7554/eLife.27702.011

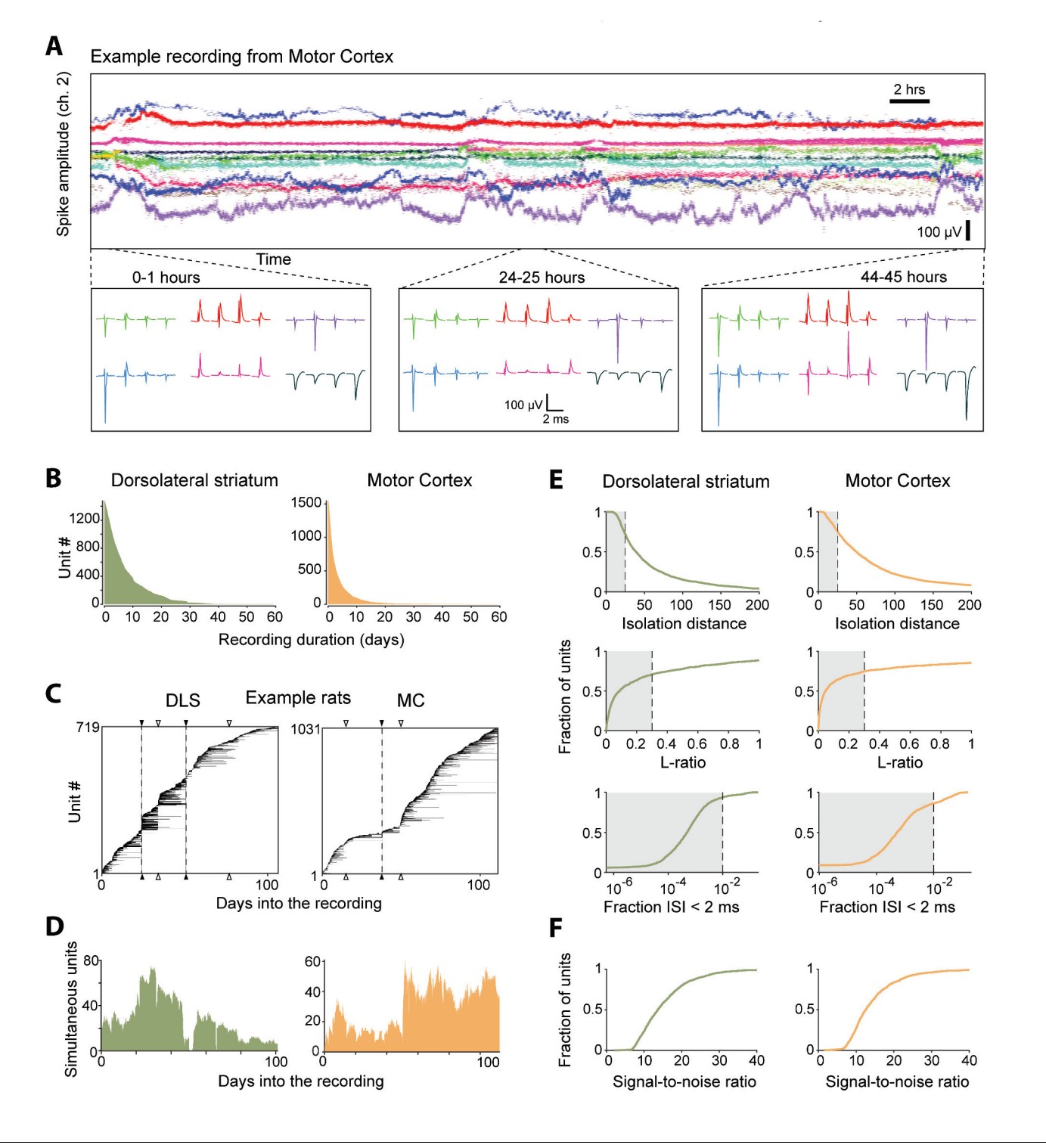

**Figure 4.** Single units isolated and tracked from continuous months-long recordings made in dorsolateral striatum (DLS) and motor cortex (MC) of behaving rats. (**A**) The results of applying FAST to a representative continuous tetrode recording in the rodent motor cortex spanning 45 hr. Each point represents spike amplitudes on channel 2 of a tetrode, with different colors representing distinct units. For visual clarity we have plotted the amplitude of de-noised spike centroids (see **Figure 2**), not raw spike amplitudes. Insets (bottom) show average spike waveforms of six example units on the four channels of the tetrode at 0, 24 and 44 hr into the recording. (**B**) Holding times for all units recorded in the DLS (left, green) and MC (right, orange), sorted by duration. (**C**) Temporal profile of units recorded in DLS (left) and MC (right) from two example rats over a period of ~3 months. Unit recording times are indicated by black bars, and are sorted by when they were first identified in the recording. Black triangles and dotted lines indicate times at

*Figure 4 continued on next page*

*Figure 4 continued*

which the electrode array was intentionally advanced into the brain by turning the micro-drive. Open triangles indicate times at which the population of recorded units changed spontaneously. Number of simultaneously recorded units in the DLS (left, green) and MC (right, orange) as a function of time in the two example recordings. (D) Number of simultaneously recorded units in the DLS (left, green) and MC (right, orange) as a function of time in the example recordings shown in C. (E) Cumulative distributions of average cluster isolation quality for all units recorded in DLS (left, green) and MC (right, orange). Cluster quality was measured by the isolation distance (top), L-ratio (middle), and fraction of inter-spike intervals under 2 ms (bottom). Dotted lines mark the quality thresholds for each of these measures. Shaded regions denote acceptable values. (F) Cumulative distributions of average signal-to-noise ratios for all units recorded in DLS (left, green) and MC (right, orange).

DOI: https://doi.org/10.7554/eLife.27702.012
The following figure supplement is available for figure 4:

**Figure supplement 1.** Benchmarking the performance of discontinuous unit-tracking methods using continuous long-term datasets sorted by FAST.
DOI: https://doi.org/10.7554/eLife.27702.013

cortex and $41 \pm 20$ units in striatum (mean $\pm$ standard deviation) (two example rats shown in *Figure 4D*).

The quality of single unit isolation was assessed by computing quantitative measures of cluster quality, that is, cluster isolation distance (*Harris et al., 2000*), L-ratio (*Schmitzer-Torbert et al., 2005*), and presence of a clean refractory period (*Hill et al., 2011*; *Lewicki, 1998*) (*Figure 4E*). Assessing the mean cluster quality of the units over their recording lifetimes, we found that 61.4% of motor cortical (n = 952) and 64.6% of striatal units (n = 954) satisfied our conservative criteria (*Quirk et al., 2009*; *Schmitzer-Torbert et al., 2005*; *Sigurdsson et al., 2010*) (Isolation distance >= 25, L-ratio <= 0.3 and fraction of ISIs below 2 ms <= 1%). However, 83.7% of motor cortical units (n = 1298) and 93.2% of striatal units (n = 1376) met these criteria for at least one hour of recording. The average signal-to-noise ratio of units isolated from the DLS and MC was found to be $15.4 \pm 7.4$ and $14.5 \pm 7.2$, respectively (*Figure 4F*).

Previous attempts to track populations of extracellularly recorded units over time have relied on matching units across discontinuous recording sessions based on similarity metrics such as spike waveform distance (*Dickey et al., 2009*; *Emondi et al., 2004*; *Fraser and Schwartz, 2012*; *Ganguly and Carmena, 2009*; *Greenberg and Wilson, 2004*; *Jackson and Fetz, 2007*; *McMahon et al., 2014a*; *Okun et al., 2016*; *Thompson and Best, 1990*; *Tolias et al., 2007*) and activity measures including firing rates and inter-spike interval histograms (*Dickey et al., 2009*; *Fraser and Schwartz, 2012*). Given that spike waveforms of single-units can undergo significant changes even within a day (*Figure 1C,D*), a major concern with discontinuous tracking methods is the difficulty in verifying their performance. Since we were able to reliably track units continuously over weeks, we used the sorted single-units as 'ground truth' data with which to benchmark the performance of day-by-day discontinuous tracking methods (*Figure 4—figure supplement 1*). In our data set, we found discontinuous tracking over day-long gaps to be highly error-prone (*Figure 4—figure supplement 1B–C*). When applying tolerant thresholds for tracking units across time, a large fraction of distinct units were labeled as the same (false positives), while at conservative thresholds only a small proportion of the units were reliably tracked across days (*Figure 4—figure supplement 1C*).

The ability to record and analyze the activity of large populations of single units continuously over days and weeks allows neural processes that occur over long timescales to be interrogated. Below we analyze and describe single neuron activity, interneuronal correlations, and the relationship between neuronal dynamics and different behavioral states, over weeks-long timescales.

## Stability of neural activity

We clustered motor cortical and striatal units into putative principal neurons and interneurons based on their spike shapes and firing rates (*Barthó et al., 2004*; *Berke et al., 2004*; *Connors and Gutnick, 1990*), thus identifying 366 fast spiking and 1111 medium spiny neurons in striatum, and 686 fast spiking and 864 regular spiking neurons in motor cortex (*Figure 5A*). Consistent with previous reports (*Hromádka et al., 2008*; *Mizuseki and Buzsáki, 2013*), average firing rates appeared lognormally distributed and varied over more than three orders of magnitude (from 0.029 Hz to 40.2 Hz in striatum, and 0.031 Hz to 44.8 Hz in motor cortex; *Figure 5A*).

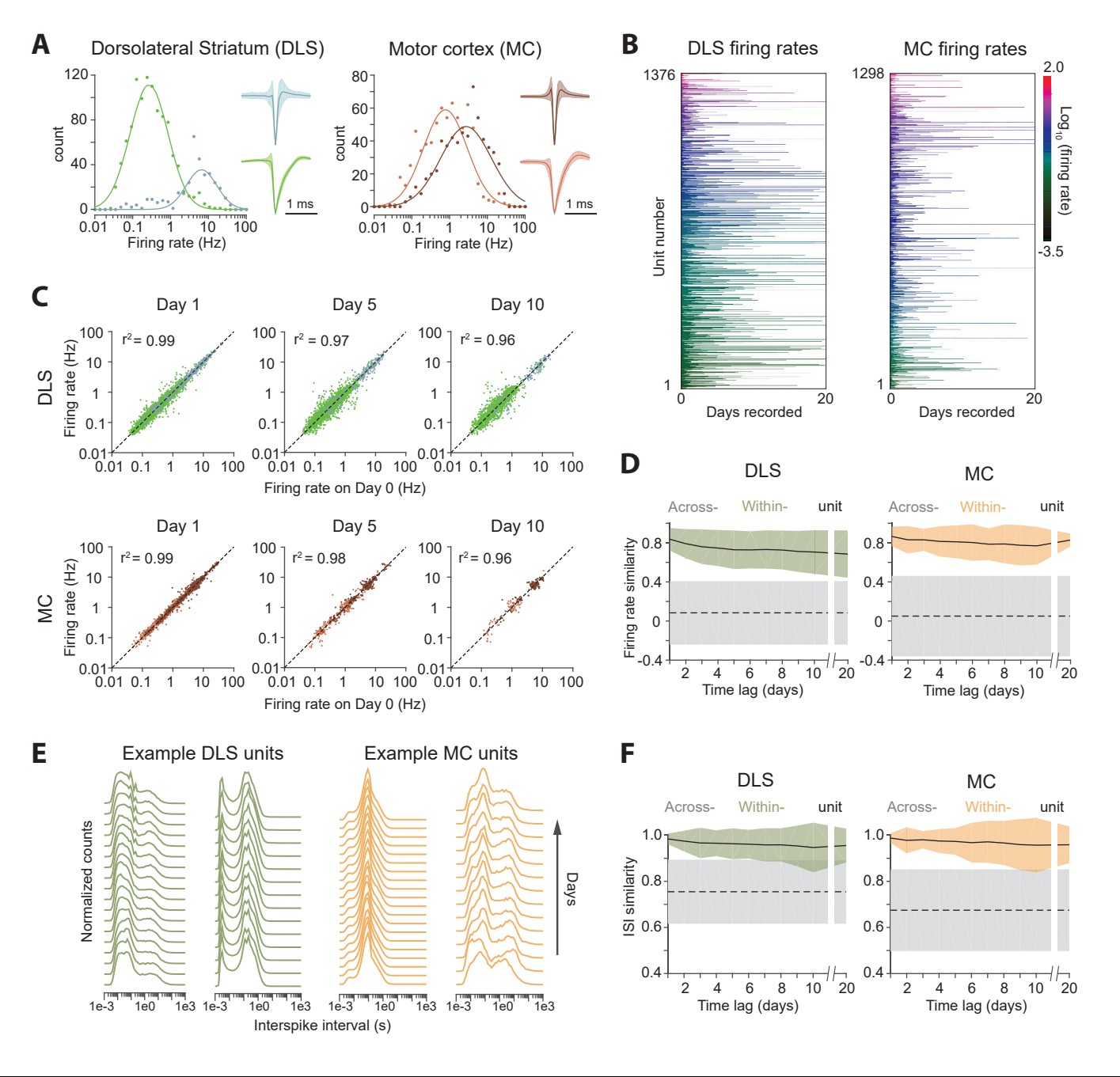

**Figure 5.** Long-term stability of single unit activity. (**A**) Histograms of average firing rates for units recorded in DLS (left) and MC (right). Putative cell-types, medium spiny neurons (MSN, blue) and fast-spiking interneurons (FSI, green) in DLS, and regular spiking (RS, brown) and fast spiking (FS, red) neurons in the MC, were classified based on spike shape and firing rate (Materials and methods). The continuous traces are log-normal fits to the firing rate distributions of each putative cell-type. Insets show average peak-normalized waveform shapes for MSNs (left-bottom) and FSIs (left-top), and RS (right-bottom) and FS (right-top) neurons. Shading represents the standard deviation of the spike waveforms. (**B**) Firing rates of DLS (left) and MC (right) units over 20 days of recording. The color scale indicates firing rate on a log-scale, calculated in one-hour blocks. Units have been sorted by average firing rate. (**C**) Scatter plots of unit firing rates over time-lags of 1 (left), 5 (middle) and 10 (right) days for DLS (top) and MC (bottom). The dashed lines indicate equality. Every dot is a comparison of a unit's firing from a baseline day to 1, 5 or 10 days later. The color of the dot indicates putative cell-type as per (**A**). Each unit may contribute multiple data points, depending on the length of the recording. Day 1: n = 4398 comparisons for striatum and n = 1458 for cortex; Day 5: n = 2471 comparisons for striatum and n = 615 for cortex; Day 10: n = 1347 comparisons for striatum and n = 268 for cortex. (**D**) Stability of unit firing rates over time. The firing rate similarity (see Materials and methods) was measured across time-lags of 1 to 20 days for the same unit (within-unit, solid lines), or between simultaneously recorded units (across-unit, dashed lines) in DLS (left) and MC (right). Colored shaded

*Figure 5 continued on next page*

*Figure 5 continued*

regions indicate the standard deviation of within-unit firing rate similarity, over all units. Grey shaded regions indicate standard deviation of across-unit firing rate similarity, over all time-bins. (E) Inter-spike interval (ISI) histograms for example units in DLS (left, green) and MC (right, orange) over two weeks of continuous recordings. Each line represents the normalized ISI histogram measured on a particular day. (F) Stability of unit ISI distributions over time. Correlations between ISI distributions were measured across time-lags of 1 to 20 days for the same unit (within-unit, solid lines), or between simultaneously recorded units (across-unit, dashed lines) in DLS (left) and MC (right). Colored shaded regions indicate the standard deviation of within-unit ISI similarity, over all units. Grey shaded regions indicate standard deviation of across-unit ISI similarity, over all time-bins.
DOI: https://doi.org/10.7554/eLife.27702.014

The following figure supplement is available for figure 5:

**Figure supplement 1.** Stability over time of ISI distributions computed from spike chains that were automatically identified by FAST.
DOI: https://doi.org/10.7554/eLife.27702.015

Average firing rates remained largely unchanged even over 20 days of recording (*Figure 5B–D*), suggesting that activity levels of single units in both cortex and striatum are stable over long time-scales, and that individual units maintain their own firing rate set-point (*Hengen et al., 2016*; *Marder and Goaillard, 2006*). We next asked whether neurons also maintain second-order spiking statistics. The inter-spike interval (ISI) distribution is a popular metric that is sensitive to a cell's mode of spiking (bursting, tonic etc.) and other intrinsic properties, such as refractoriness and spike frequency adaptation. Similar to firing rate, we found that the ISI distribution of single units remained largely unchanged across days (*Figure 5E–F*). To verify whether our ISI similarity criterion for manual merging of FAST-generated spike chains (see Materials and methods) biased our measurements of ISI stability, we also analyzed the stability of the ISI computed from chains generated by the fully automated steps 1 and 2 of FAST, and found it to be similarly stable over multiple days (*Figure 5— figure supplement 1*).

Measures of single unit activity do not adequately address the stability of the network in which the neurons are embedded, as they do not account for interneuronal correlations (*Abbott and Dayan, 1999*; *Ecker et al., 2010*; *Nienborg et al., 2012*; *Okun et al., 2015*; *Salinas and Sejnowski, 2001*; *Singer, 1999*). To address this, we calculated the cross-correlograms of all simultaneously recorded neuron pairs (*Figure 6A*, Materials and methods). We found that 21.1% of striatal pairs (n = 18697 pairs) and 37.6% motor cortex pairs (n = 9235 pairs) were significantly correlated (Materials and methods). The average absolute time-lag of significantly correlated pairs was 13.0 ± 26.3 ms (median: 4 ms) and 23.9 ± 28.8 ms (median: 13.0 ms) for striatum and motor cortex respectively. The pairwise spike correlations were remarkably stable, remaining essentially unchanged even after 10 days (*Figure 6B–C*), consistent with a very stable underlying network (*Grutzendler et al., 2002*; *Yang et al., 2009*).

## Stability of behavioral state-dependent activity

Our results demonstrated long-term stability in the time-averaged statistics of single units (*Figure 5*) as well as their interactions (*Figure 6*), both in motor cortex and striatum. However, the brain functions to control behavior, and the activity of both cortical and striatal units can be expected to differ for different behaviors. To better understand the relationship between the firing properties of single units and ongoing behavior, and to assess the stability of these relationships, we analyzed single unit activity in different behavioral states (*Figure 7*). To do this, we developed an algorithm for automatically classifying a rat's behavior into 'states' using high-resolution measurements of head acceleration and local field potentials (LFPs). Our method distinguishes repetitive behaviors (such as grooming), eating, active exploration, task engagement as well as quiet wakefulness, rapid eye movement, and slow wave sleep (*Gervasoni et al., 2004*; *Venkatraman et al., 2010*) (*Figure 7—figure supplement 1*, Materials and methods), and assigns ~88% of the recording time to one of these states.

Behavioral state transitions, such as between sleep and wakefulness, have been shown to affect average firing rates (*Evarts, 1964*; *Lee and Dan, 2012*; *Peña et al., 1999*; *Santhanam et al., 2007*; *Steriade et al., 1974*; *Vyazovskiy et al., 2009*), inter-spike interval distributions (*Mahon et al., 2006*; *Vyazovskiy et al., 2009*), and neural synchrony (*Gervasoni et al., 2004*; *Ribeiro et al., 2004*; *Wilson and McNaughton, 1994*). However, most prior studies analyzed brief recording sessions in which only a narrow range of behavioral states could be sampled (*Hirase et al., 2001*; *Mahon et al.,*

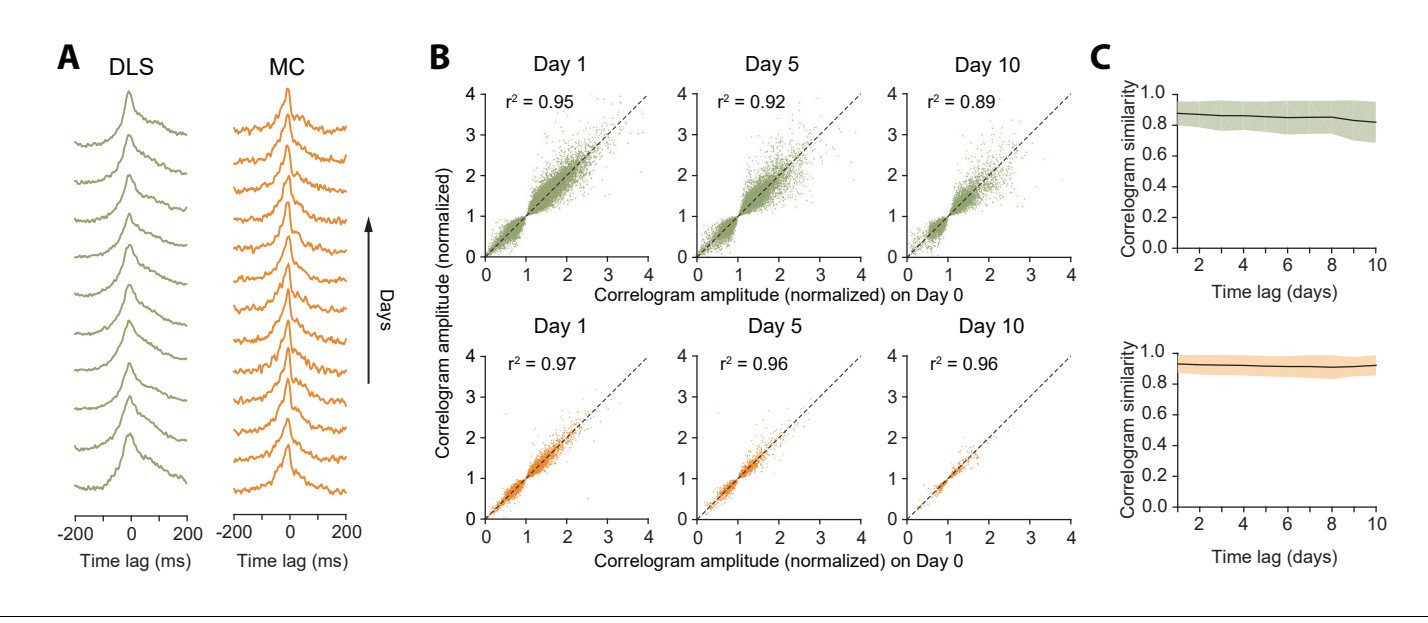

**Figure 6.** Stability of network dynamics. (**A**) Correlograms for example unit pairs from DLS (left) and MC (right) over 10+ days. Each line represents the normalized correlogram measured on a particular day. (**B**) Scatter plots comparing the correlogram amplitude (normalized) of a unit pair on a baseline day to the same pair's peak correlation 1, 5 or 10 days later, for DLS (top) and MC (bottom). The black line indicates equality. Values > 1 (or <1) correspond to positive (or negative) correlations (see Materials and methods). Day 1: n = 48472 comparisons for DLS and n = 12148 for MC. Day 5: n = 19230 comparisons for DLS and n = 2881 for MC. Day 10: n = 7447 comparisons for DLS and n = 772 for MC. All unit pairs whose correlograms had at least 5000 spikes over a day's period were included. (**C**) Stability of correlations over time. The correlogram similarity (see Materials and methods) was measured across time-lags of 1 to 10 days for pairs of units (solid lines) in DLS (top) and MC (bottom). Colored shaded regions indicate the standard deviation of the correlogram similarity between all pairs that had significant correlograms on at least one recording day (Materials and methods).

DOI: https://doi.org/10.7554/eLife.27702.016

2006; *Vyazovskiy et al., 2009*; *Wilson and McNaughton, 1994*), leaving open the question of whether the modulation of neural dynamics across behavioral states is distinct for different neurons in a network and whether state-specific activity patterns of single neurons are stable over time. Not surprisingly, we found that firing rates of both striatal and motor cortical units depended on what the animal was doing (*Figure 7A*). Interestingly, the relative firing rates in different behavioral states (i.e. a cell's 'state tuning') remained stable over several weeks-long timescales (*Figure 7A–B*), even as they varied substantially between different neurons (*Figure 7A,C*). Additionally, we observed that interneuronal spike correlations were also modulated by behavioral state (*Figure 7D*), and this dependence was also stable over time (*Figure 7E*).

## Stability in the coupling between neural and behavioral dynamics

Neurons in motor-related areas encode and generate motor output, yet the degree to which the relationship between single unit activity and various movement parameters (i.e. a cell's 'motor tuning') is stable over days and weeks has been the subject of recent debate (*Carmena et al., 2005*; *Chestek et al., 2007*; *Clopath et al., 2017*; *Ganguly and Carmena, 2009*; *Huber et al., 2012*; *Liberti et al., 2016*; *Peters et al., 2014*; *Rokni et al., 2007*; *Stevenson et al., 2011*). Long-term continuous recordings during natural and learned behaviors constitute unique datasets for characterizing the stability of movement coding at the level of single neurons.

We addressed this by computing the 'response fields' (*Cheney and Fetz, 1984*; *Cullen et al., 1993*; *Serruya et al., 2002*; *Wessberg et al., 2000*), i.e. the spike-triggered average (STA) accelerometer power (Materials and methods), of neurons in different active states (*Figure 8A*; Materials and methods). The percentage of units with significant response fields during repetitive behavior, exploration, and eating was 19.7% (n = 291), 9.5% (n = 140) and 18.4% (n = 272), respectively, for the striatum, and 18.7% (n = 290), 7.6% (n = 118) and 22.5% (n = 349) for the motor cortex.

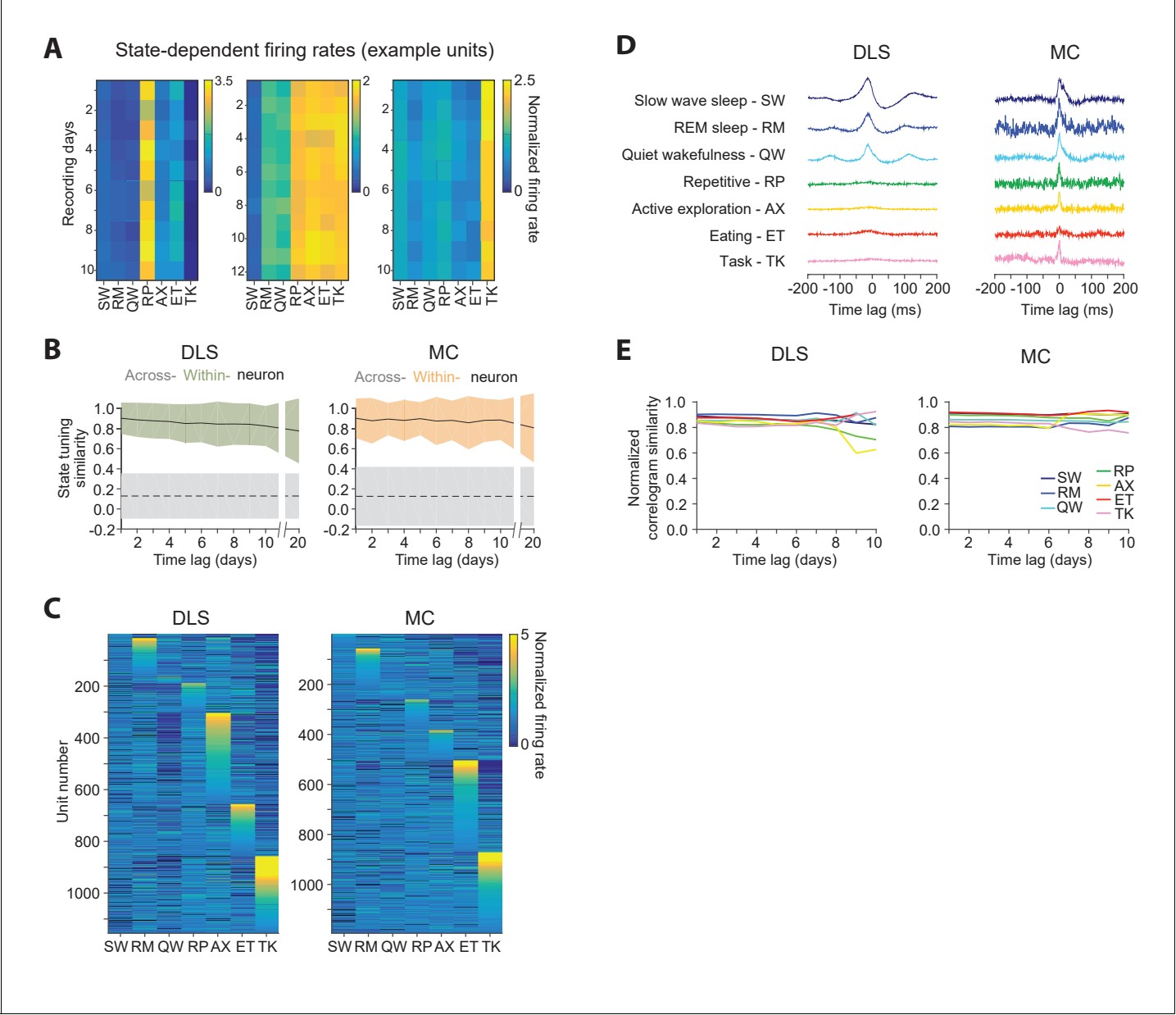

**Figure 7.** Stability of behavioral state-dependent activity patterns. (**A**) Stability of average firing rates in different behavioral states (i.e. the unit's 'state-tuning') across several days for example units recorded in the DLS (left) and MC (middle and right). The color-scale corresponds to the firing rate of a unit normalized by its average firing rate. The state abbreviations correspond to slow-wave sleep (SW), REM sleep (RM), quiet wakefulness (QW), repetitive (RP), active exploration (AX), eating (ET) and task execution (TK). (**B**) Stability of units' state-tuning over time. Correlations between the state-tuning profiles (as in 'A') were measured across time-lags of 1 to 20 days for the same unit (within-unit, solid lines), or between simultaneously recorded units (across-unit, dashed lines) in MC (left) and DLS (right). Colored shaded regions indicate the standard deviation of within-unit state-tuning similarity, over all units. Grey shaded regions indicate standard deviation of across-unit state-tuning similarity, over all time-bins. (**C**) Diversity in state-tuning across all units recorded in the DLS (left) and MC (right). Units are sorted by the peak behavioral state in their state-tuning curves. (**D**) Cross-correlograms computed for example pairs of units in the DLS (left) and MC (right) in different behavioral states. Line colors represent behavioral state. (**E**) Stability of cross-correlograms measured within specific behavioral states. The correlation similarity (see Materials and methods) was measured across time-lags of 1 to 10 days (solid lines) in DLS (left) and MC (right). Line colors represent correlation similarity for different behavioral states, as per the plot legend.

DOI: https://doi.org/10.7554/eLife.27702.017

The following figure supplement is available for figure 7:

**Figure supplement 1.** Automated classification of behavioral states.
DOI: https://doi.org/10.7554/eLife.27702.018

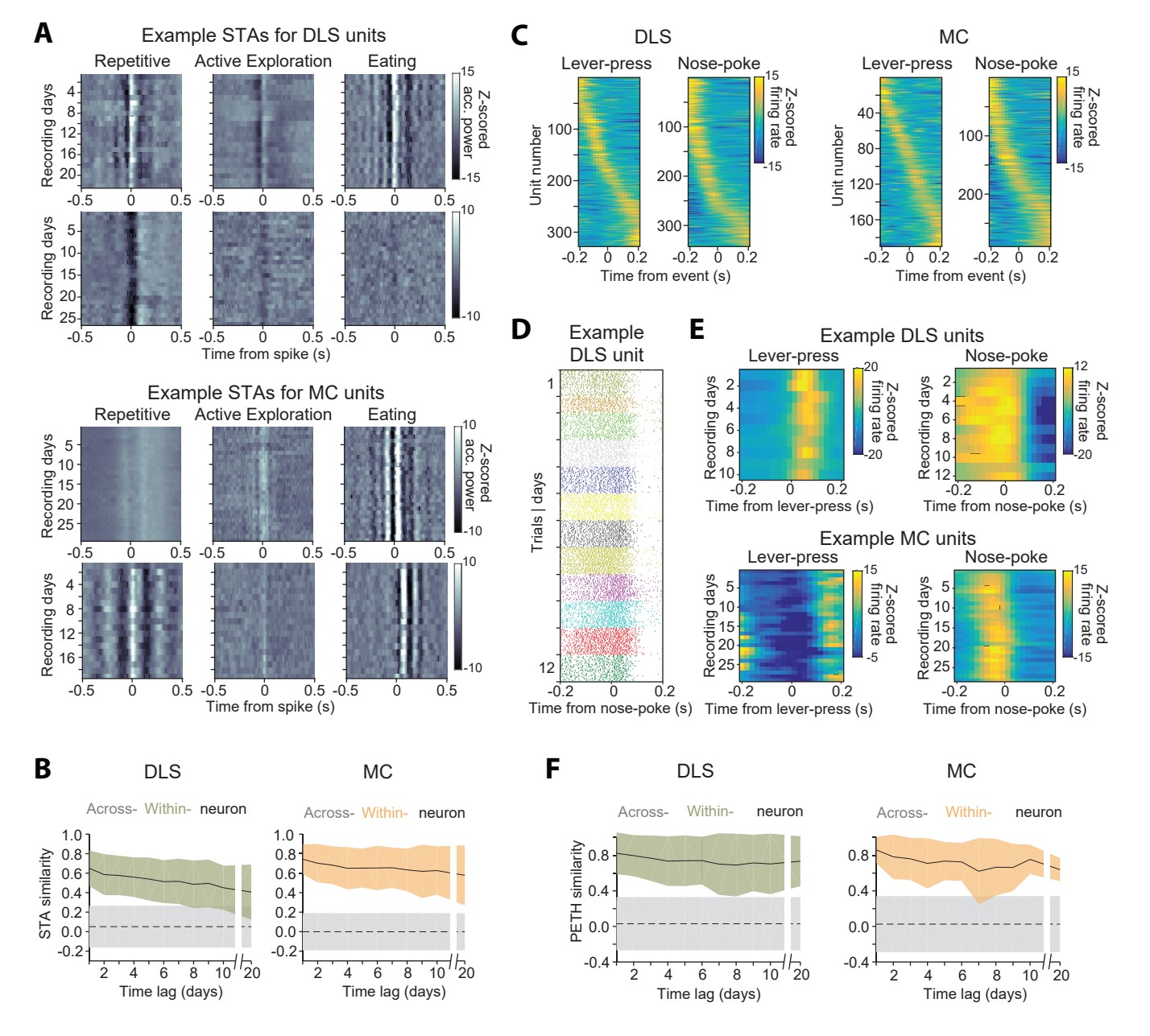

**Figure 8.** Stability of behavioral representations. (**A**) Spike-triggered average (STA) accelerometer power calculated daily in three different behavioral states – repetitive behavior (left), active exploration (middle) and eating (right). Shown are four example units recorded from DLS (top two rows) and MC (bottom two rows). (**B**) Stability of STAs over time. Correlations between STAs were measured across time-lags of 1 to 20 days for the same unit (within-unit, solid lines), or between simultaneously recorded units (across-unit, dashed lines) in DLS (left) and MC (right). Colored shaded regions indicate the standard deviation of within-unit STA similarity (averaged over the three behavioral states in 'A'), over all units. Grey shaded regions indicate standard deviation of across-unit STA similarity, over all time-bins. (**C**) Peri-event time histograms (PETHs) of DLS (left) and MC (right) unit activity, aligned to the timing of a lever-press or nose-poke during the execution of a skilled motor task. Plotted are the PETHs of units that had significant modulations in their firing rate in a time-window ±200 ms around the time of the behavioral event (Materials and methods). The color scale indicates Z-scored firing rate. Units are sorted based on the times of peaks in their PETHs. (**D**) Spike raster of an example DLS unit over 12 days, aligned to the time of a nose-poke event. Each dot represents a spike-time on a particular trial. The color of the dot indicates day of recording. (**E**) PETHs computed over several days for example DLS (top) and MC (bottom) units to lever-press (left) and nose-poke (right) events in our task. (**F**) Stability of task PETHs over time. Correlations between PETHs were measured across time-lags of 1 to 20 days for the same unit (within-unit, solid lines), or between simultaneously recorded units (across-unit, dashed lines) in DLS (left) and MC (right). Colored shaded regions indicate the standard deviation

*Figure 8 continued*

of within-unit PETH similarity (averages across lever-press and nose-poke events), over all units. Grey shaded regions indicate standard deviation of across-unit PETH similarity (averaged across lever-press and nose-poke events), over all time-bins.

DOI: https://doi.org/10.7554/eLife.27702.019

The following figure supplement is available for figure 8:

**Figure supplement 1.** Measurements of long-term stability of neural dynamics and motor representations for individual units.

DOI: https://doi.org/10.7554/eLife.27702.020

Movement STAs varied substantially between neurons and across behavioral states, but were stable for the same unit when compared over at least 20 days, within a particular state (*Figure 8A–B*).

Another measure used to characterize neural coding of behavior on fast timescales is the 'peri-event time histogram' (PETH), that is, the average neural activity around the time of a salient behavioral event. We calculated PETHs for two events reliably expressed during the execution of a skill our subjects performed in daily training sessions (*Kawai et al., 2015*) (Materials and methods): (i) the first lever-press in a learned lever-press sequence and (ii) entry into the reward port after successful trials ('nose-poke'). The fraction of units whose activity was significantly locked to the lever-press or nose-poke (see Materials and methods) was 22.2% (n = 328) and 23.2% (n = 343) in striatum, and 12.3% (n = 190) and 18.7% (n = 290) in motor cortex (*Figure 8C*), respectively. When comparing PETHs of different units for the same behavioral events, we observed that the times of peak firing were distributed over a range of delays relative to the time of the events (*Figure 8C*). However, despite the heterogeneity in PETHs across the population of units, the PETHs of individual units were remarkably similar when compared across days (*Figure 8D–F*).

## Stability of individual units

Our population-level analyses showed that many aspects of neural activity, such as firing rates, inter-spike intervals and behavioral representations are stable over many weeks. While these results are consistent with the majority of recorded units being stable over these timescales, they do not rule out the existence of a smaller subpopulation of units whose dynamics may be unstable over time. To probe this, we did a regression analysis to measure the dependence of different similarity metrics on elapsed time (*Figure 8—figure supplement 1A*), and quantified the stability index of individual units as the slope of the best-fit line. The distribution of stability indices was unimodal and centered at 0 $day^{-1}$ for every similarity metric, indicating that the vast majority (between 88% and 98%) of recorded units were stable (*Figure 8—figure supplement 1B*, *Table 2*) given a conservative threshold (slope >= -0.025 $day^{-1}$ and p>=0.05). Interestingly, a negative skew was also apparent in these distributions implying that a small fraction (~2–12%) of our recorded units were relatively less stable over longer timescales (*Figure 8—figure supplement 1B*, *Table 2*); however no unit was found to have a stability index lower than -0.1 $day^{-1}$.

**Table 2.** Proportion of units that have stable neural dynamics and motor representations in the striatum and motor cortex.

| Activity metric | Dorsolateral striatum | | Motor cortex | |
|---|---|---|---|---|
| | % stable | n | % stable | n |
| Firing rate | 90.3 | 186 | 88.1 | 42 |
| ISI histogram | 97.9 | 187 | 97.6 | 42 |
| State tuning | 93.9 | 163 | 92.7 | 41 |
| STA | 93.6 | 202 | 95.0 | 80 |
| PETH | 89.4 | 113 | 89.5 | 57 |

DOI: https://doi.org/10.7554/eLife.27702.021

## Discussion

Recording from populations of single neurons in behaving animals has traditionally been a laborious undertaking both in terms of experimentation and analysis. Automating this process, and extending the recordings over weeks and months, increases the efficiency and power of such experiments in multiple ways. First, it eliminates the manual steps of intermittent recordings, such as transferring animals to and from their recording chambers, plugging and unplugging recording cables etc., which, besides being time-consuming, can be detrimental to the recordings (*Buzsáki et al., 2015*) and stressful for the animals (*Dobrakovová and Jurcovicová, 1984*; *Longordo et al., 2011*). Second, when combined with fully automated home-cage training (*Poddar et al., 2013*), this approach allows the neural correlates of weeks- and months-long learning processes to be studied in an efficient manner. Indeed, our system should enable a single researcher to supervise tens, if not hundreds, of recordings simultaneously, thus radically improving the ease and throughput with which such experiments can be performed. Third, continuous recordings allow the activity of the same neurons to be tracked over days and weeks, allowing their activity patterns to be pooled and compared for more trials, and across different experimental conditions and behavioral states, thus increasing the power with which the principles of neural function can be inferred (*Lütcke et al., 2013*; *McMahon et al., 2014a*).

Despite their many advantages, continuous long-term neural recordings are rarely performed. This is in large part because state-of-the-art methods for processing extracellular recordings require significant human input and hence make the analysis of large datasets prohibitively time-consuming (*Hengen et al., 2016*). Our fast automated spike tracking (FAST) algorithm overcomes this bottleneck, thereby dramatically increasing the feasibility of continuous and high-throughput long-term recordings of extracellular signals in behaving animals.

### Long-term stability of neural dynamics

Recording the activity of the same neuronal population over long time periods can be done by means of functional calcium imaging (*Huber et al., 2012*; *Lütcke et al., 2013*; *Peters et al., 2014*; *Rose et al., 2016*). However, the calcium signal reports a relatively slow correlate of neural activity that can make it difficult to reliably resolve individual spike events and hence to assess fine timescale neuronal interactions (*Grienberger and Konnerth, 2012*; *Lütcke et al., 2013*). Furthermore, calcium indicators function as chelators of free calcium ions (*Hires et al., 2008*; *Tsien, 1980*) and could interfere with calcium-dependent plasticity processes (*Zucker, 1999*) and, over the long-term, compromise the health of cells (*Looger and Griesbeck, 2012*). These variables may have contributed to discrepancies in studies using longitudinal calcium imaging, with many reporting dramatic day-to-day fluctuations in neural activity patterns (*Driscoll et al., 2017*; *Huber et al., 2012*; *Liberti et al., 2016*; *Ziv et al., 2013*) while some others report more stable representations (*Peters et al., 2014*; *Rose et al., 2016*), leaving open the question of how stable the brain really is (*Clopath et al., 2017*).

While electrical measurements of spiking activity do not suffer from the same drawbacks, intermittent recordings may fail to reliably track the same neurons over multiple days (*Figure 3—figure supplement 1D–E*) (*Dickey et al., 2009*; *Emondi et al., 2004*; *Fraser and Schwartz, 2012*; *Tolias et al., 2007*). Attempts at inferring long-term stability of movement-related neural dynamics from such experiments have produced conflicting results, with many studies reporting stability (*Bondar et al., 2009*; *Chestek et al., 2007*; *Fraser and Schwartz, 2012*; *Ganguly and Carmena, 2009*; *Greenberg and Wilson, 2004*; *Nuyujukian et al., 2014*), while others find significant fluctuations even within a single day (*Carmena et al., 2005*; *Mankin et al., 2012*; *Perge et al., 2013*; *Rokni et al., 2007*). Such contrasting findings could be due to systematic errors in unit tracking across discontinuous recording sessions (*Figure 4—figure supplement 1*), or the choice of statistical framework used to assess stability (*Rokni et al., 2007*; *Stevenson et al., 2011*). Complications may also arise from the use of activity metrics, such as firing rate and ISI distributions, to track units in many studies (*Dickey et al., 2009*; *Emondi et al., 2004*; *Fraser and Schwartz, 2012*; *Ganguly and Carmena, 2009*; *Greenberg and Wilson, 2004*; *Jackson and Fetz, 2007*; *McMahon et al., 2014a*; *Okun et al., 2016*; *Thompson and Best, 1990*; *Tolias et al., 2007*), which may bias datasets towards subsets of units whose spike waveforms (*Figure 1C–D*) and neural dynamics (*Figure 8—figure supplement 1*) do not change with time.

Our continuous recordings, which allow for the reliable characterization of single neuron properties over long time periods, suggest a remarkable stability in both spiking statistics (*Figure 5*), neuronal interactions (*Figure 6*), and tuning properties (*Figures 7* and *8*) of the majority of motor cortical and striatal units. Taken together with studies showing long-term structural stability at the level of synapses (*Grutzendler et al., 2002*; *Yang et al., 2009*), our findings support the view that these neural networks are, overall, highly stable systems, both in terms of their dynamics and connectivity.

Discrepancies between our findings and other studies that report long-term instability of neural activity could, in some cases, be due to inherent differences between brain areas. Neural dynamics of circuits involved in transient information storage such as the hippocampus (*Mankin et al., 2012*; *Ziv et al., 2013*), and flexible sensorimotor associations such as the parietal cortex (*Morcos and Harvey, 2016*) may be intrinsically more unstable than circuits responsible for sensory processing or motor control (*Ganguly and Carmena, 2009*; *McMahon et al., 2014b*; *Peters et al., 2014*; *Rose et al., 2016*). In line with this hypothesis, recent studies have found higher synaptic turnover in the hippocampus (*Attardo et al., 2015*) in comparison to primary sensory and motor cortices (*Holtmaat et al., 2005*; *Xu et al., 2009*; *Yang et al., 2009*), although it is not clear whether structural plasticity functions to destabilize or stabilize neural activity (*Clopath et al., 2017*). In this context, our long-term electrical recording and analysis platform could be used to verify whether spiking activity in these associative areas is as unstable as their reported calcium dynamics.

In this study we recorded from rats that had already learned a sequence task in order to answer the question: is the brain stable when behavior is stable? We hypothesize that neural activity would be far more variable when an animal is learning a task or motor skill (*Kawai et al., 2015*). Our experimental platform makes it feasible to characterize learning-related changes in task representations and relate them to reorganization of network dynamics over long time-scales.

## Future improvements

There is a continued push to improve recording technology to increase the number of neurons that can be simultaneously recorded from (*Berényi et al., 2014*; *Buzsáki et al., 2015*; *Du et al., 2011*; *Viventi et al., 2011*), as well as make the recordings more stable (*Alivisatos et al., 2013*; *Chestek et al., 2011*; *Cogan, 2008*; *Fu et al., 2016*; *Guitchounts et al., 2013*). Our modular and flexible experimental platform can easily incorporate such innovations. Importantly, our novel and automated analysis pipeline (FAST) solves a major bottleneck downstream of these solutions by allowing increasingly large datasets to be efficiently parsed, thus making automated recording and analysis of large populations of neurons in behaving animals a feasible prospect.

## Materials and methods

### Animals

The care and experimental manipulation of all animals were reviewed and approved by the Harvard Institutional Animal Care and Use Committee. Experimental subjects were female Long Evans rats (RRID:RGD_2308852), 3–8 months old at the start of the experiment (n = 5, Charles River).

### Behavioral training

Before implantation, rats were trained twice daily on a timed lever-pressing task (*Kawai et al., 2015*) using our fully automated home-cage training system (*Poddar et al., 2013*). Once animals reached asymptotic (expert) performance on the task, they underwent surgery to implant tetrode drives into dorsolateral striatum (n = 2) and motor cortex (n = 3) respectively. After 7 days of recovery, rats were returned to their home-cages, which had been outfitted with an electrophysiology recording extension (*Figure 1A*, *Figure 1—figure supplement 1*). The cage was placed in an acoustic isolation box, and training on the task resumed. Neural and behavioral data was recorded continuously for 12–16 weeks with only brief interruptions (median time of 0.2 hr) for occasional troubleshooting.

## Continuous behavioral monitoring

To monitor the rats' head movements continuously, we placed a small 3-axis accelerometer (ADXL 335, Analog Devices) on the recording head-stage (*Figure 1—figure supplement 1*). The output of the accelerometer was sampled at 7.5 kHz per axis. We also recorded 24/7 continuous video at 30 frames per second with a CCD camera (Flea 3, Point Grey) or a webcam (Agama V-1325R). Video was synchronized to electrophysiological signals by recording TTL pulses from the CCD cameras that signaled frame capture times.

## Surgery

Rats underwent surgery to implant a custom-built recording device (see 'Tetrode arrays'). Animals were anesthetized with 1–3% isoflurane and placed in a stereotax. The skin was removed to expose the skull and five bone screws (MD-1310, BASi), including one soldered to a 200 µm diameter silver ground wire (786500, A-M Systems), were driven into the skull to anchor the implant. A stainless-steel reference wire (50 µm diameter, 790700, AM-Systems) was implanted in the external capsule to a depth of 2.5 mm, at a location posterior and contralateral to the implant site of the electrode array. A 4–5 mm diameter craniotomy was made at a location 2 mm anterior and 3 mm lateral to bregma for targeting electrodes to motor cortex, and 0.5 mm anterior and 4 mm lateral to bregma for targeting dorsolateral striatum. After removing the dura-mater, the pia-mater surrounding the implant site was glued to the skull with cyanoacrylate glue (Krazy glue). The pia-mater was then weakened using a solution of 20 mg/ml collagenase (Sigma) and 0.36 mM calcium chloride in 50 mM HEPES buffer (pH 7.4) in order to minimize dimpling of the brain surface during electrode penetration (*Kralik et al., 2001*). The 16-tetrode array was then slowly lowered to the desired depth of 1.85 mm for motor cortex and 4.5 mm for striatum. The microdrive was encased in a protective shield and cemented to the skull by applying a base layer of Metabond luting cement (Parkell) followed by a layer of dental acrylic (A-M systems).

## Histology

At the end of the experiments, animals were anesthetized and anodal current (30 µA for 30 s) passed through select electrodes to create micro-lesions at the electrode tips. Animals were transcardially perfused with phosphate-buffered saline (PBS) and subsequently fixed with 4% paraformaldehyde (PFA, Electron Microscopy Sciences) in PBS. Brains were removed and post-fixed in 4% PFA. Coronal sections (60 µm) were cut on a Vibratome (Leica), mounted, and stained with cresyl violet to reconstruct the location of implanted electrodes. All electrode tracks were consistent with the recordings having been done in the targeted areas.

## Tetrode arrays

Tetrodes were fabricated by twisting together short lengths of four 12.5 µm diameter nichrome wires (Redi Ohm 800, Sandvik-Kanthal), after which they were bound together by melting their polyimide insulation with a heat-gun. An array of 16 such tetrodes was attached to a custom-built microdrive, advanced by a 0–80 threaded screw (~320 µm per full turn). The wires were electroplated in a gold (5355, SIFCO) and 0.1% carbon nanotubes (Cheap Tubes dispersed MWNTs, 95wt% <8 nm) solution with 0.1% polyvinylpyrrolidone surfactant (PVP-40, Sigma-Aldrich) to lower electrode impedances to ~100–150 kΩ (*Ferguson et al., 2009*; *Keefer et al., 2008*). The prepared electrode array was then implanted into the motor cortex or striatum.

## Tethering system

We used extruded aluminum with integrated V-shaped rails for the linear-slide (Inventables, Part #25142–03), with matching wheels (with V-shaped grooves) on the carriage (Inventables, Part #25203–02). The bearings in the wheels were degreased and coated with WD-40 to minimize friction. The carriage plate was custom designed (design files available upon request) and fabricated by laser cutting 3 mm acrylic. A low-cost 15-channel commutator (SRC-022 slip-ring, Hangzhou-Prosper Ltd) was mounted onto the carriage plate. The commutator was suspended with a counterweight using low friction pulleys (Misumi Part # MBFN23-2.1). We used a 14 conductor cable (Mouser, Part # 517-3600B/14100SF) to connect our custom designed head-stage to the commutator. The outer PVC insulation of the cable bundle was removed to reduce the weight of the cable and to make it

more flexible. The same cable was used to connect the commutator to a custom designed data acquisition system. This cable needs to be as flexible as possible to minimize forces on the animal's head.

## Recording hardware

Our lightweight, low-noise and low-cost system for multi-channel extra-cellular electrophysiology (*Figure 1—figure supplement 1*) uses a 64-channel custom-designed head-stage (made of 2 RHD2132 ICs from Intan Technologies), weighs less than four grams and measures 18 mm x 28 mm in size. The head-stage filters, amplifies, and digitizes extracellular neural signals at 16 bits/sample and 30 kHz per second per channel. An FPGA module (Opal Kelly XEM6010 with Xilinx Spartan-6 FPGA) interfaces the head-stage with a computer that stores and displays the acquired data. Since the neural signals are digitized at the head-stage, the system is low-noise and can support long cable lengths. Design files and a complete parts list for all components of the system are available on request. The parts for a complete 64-channel electrophysiology extension, including head-stage, FPGA board, commutator, and pulley system, but excluding the recording computer, cost less than $1500.

## Automatic classification of behavior

We developed an unsupervised algorithm to classify behaviors based on accelerometer data, LFPs, spike times, and task event times (*Gervasoni et al., 2004*; *Venkatraman et al., 2010*). Behaviors were classified at 1 s resolution into one of the following 'states': repetitive (RP), eating (ET), active exploration (AX), task engagement (TK), quiet wakefulness (QW), rapid eye movement sleep (RM) or slow-wave sleep (SW).

### Processing of accelerometer data

We computed the $L^2$ norm of the accelerometer signal $\|acc\|$ as follows:

$$\|acc\| = \sqrt{acc_x^2 + acc_y^2 + acc_z^2} \tag{1}$$

where $acc_i$ is the accelerometer signal on channel $i$. The accelerometer norm was then band-pass filtered between 0.5 Hz and 150 Hz using a fourth order Butterworth filter. We then computed the multitaper spectrogram (0–40 Hz) of the accelerometer signal in 1 s bins using Chronux (*Mitra and Bokil, 2007*) and measured the total accelerometer power in the 2–5 Hz frequency bands. The accelerometer power had a bimodal distribution with one narrow peak at low values corresponding to immobility and a secondary broad peak corresponding to epochs of movement. Time-bins with accelerometer power within two standard deviations of the lower immobility peak were termed 'inactive', while time-bins with power greater than the 'mobility threshold' – four standard deviations above the immobility peak – were termed 'active'.

To verify the accuracy of our accelerometer-based separation of active and inactive behaviors, we calculated the percent overlap between our automated separation with that of a human observer who manually scored active and inactive periods using simultaneously recorded webcam video data (n = 29831 s video from 4 rats).

|  | Inactive (auto) | Active (auto) |
| --- | --- | --- |
| *Inactive (manual)* | 92.9% | 7.1% |
| *Active (manual)* | 3.1% | 96.9% |

### Processing of the LFP

We extracted LFPs by down-sampling the raw data 100-fold (from 30 kHz to 300 Hz) by two applications of a fourth order 5-fold decimating Chebychev filter followed by a single application of a fourth order 4-fold decimating Chebychev filter. We then computed the multitaper spectrogram (0–20 Hz) for the LFP signal recorded on each electrode with a moving window of width 5 s and step-size 1 s using Chronux (*Mitra and Bokil, 2007*), and then averaged these spectrograms across all electrodes

to yield the final LFP spectrogram. We used PCA to reduce the dimensionality of the spectrogram to 20 components and then fit a three component Gaussian mixture model to cluster the LFP spectrograms into three distinct brain states. Based on earlier studies (*Gervasoni et al., 2004*; *Lin et al., 2006*), these states were characterized by high delta (0–4 Hz) power (e.g. slow wave sleep), high type-II theta (7–9 Hz) power (e.g. REM sleep or active exploration) and high type-I theta (5–7 Hz) power (e.g. quiet wakefulness or repetitive behaviors).

### Classification of inactive states – REM and slow-wave sleep, and quiet wakefulness

Inactive states, that is, when rats are immobile, were identified as bins with total accelerometer power below the immobility threshold (see 'Processing of accelerometer data' above). Slow-wave sleep (SW) states were distinguished by high delta power (see 'Processing of the LFP' above), REM sleep states (RM) were distinguished by high type-II theta power, and quiet wakefulness states (QW) were distinguished by high type-I theta power (*Bland, 1986*; *Gervasoni et al., 2004*; *Lin et al., 2006*). Consistent with prior studies (*Gervasoni et al., 2004*), we found that rats spent most of their time sleeping (44 ± 3%) or resting (6 ± 2%) (*Figure 7—figure supplement 1B*). Sleep occurred at all hours, but was more frequent when lights were on (58 ± 7%) versus when they were off (33 ± 9%), consistent with the nocturnal habits of rats (*Figure 7—figure supplement 1C*).

### Classification of active states – repetitive behaviors, eating, task engagement and active exploration

Active states were characterized by the accelerometer power being above the mobility threshold (see 'Processing of accelerometer data' above).

*Eating* (ET) epochs were classified based on a strong, broad-band common-mode oscillatory signal on all electrodes, due to electrical interference from muscles of mastication. To extract eating epochs, we median-filtered the raw data across all channels, and computed a multitaper spectrogram between 0–40 Hz using Chronux, with window width 5 s and step-size 1 s. After reducing the dimensionality of the spectrogram to 20 by PCA, we fit a Gaussian mixture model and identified cluster(s) corresponding to the broad-band noise characteristic of eating.

*Task engagement* (TK) was identified based on lever-press event times. If less than 2 s had elapsed between presses (corresponding to the average trial length) the time between them was classified as task engagement.

Active epochs that were not classified as eating or task engagement were classified as *active exploration* (AX) or *repetitive behavior* (RP) depending on whether they had high type II or high type I theta power in their LFP spectrograms, respectively, based on prior reports (*Bland, 1986*; *Gervasoni et al., 2004*; *Sainsbury et al., 1987*).

The algorithm labeled each time bin exclusively, i.e. only one label was possible for each bin. Bins corresponding to multiple states (e.g. eating and grooming) where classified as *unlabeled* (UL), as were times when no data was available due to brief pauses in the recording (9% of time-bins unlabeled).

### Removal of seizure-like epochs

Long Evans rats are susceptible to absence seizures, characterized by strong ~8 Hz synchronous oscillations in neural activity and associated whisker twitching (*Berke et al., 2004*; *Gervasoni et al., 2004*; *Nicolelis et al., 1995*). To identify seizure-like episodes, we calculated the autocorrelation of the raw spike data on all electrodes in 2 s windows. To identify peaks associated with seizure-like states, we calculated the periodogram of each autocorrelation function and evaluated the power in the 7–9 Hz frequency range. The distribution of these values had a Gaussian shape with a non-Gaussian long tail towards high values. The threshold for seizure-like states was set to the 95th percentile of the Gaussian distribution. Using this classification, 11% of time-bins were classified as seizure (12 ± 2% for the DLS-implanted rats and 11 ± 2% for the MC-implanted rats) similar to previously published reports (*Shaw, 2004*). These epochs were removed from the behavioral state analysis.

## Data storage and computing infrastructure

### Hardware setup

Our recordings (64-channels at 30 kHz per channel) generate 1 terabyte (TB) of raw data every 2 days. To cope with these demands, we developed a low-cost and reliable high I/O bandwidth storage solution with a custom lightweight fileserver. Each storage server consists of 24 4TB spinning SATA hard disks connected in parallel to a dual socket Intel server class motherboard via the high bandwidth PCI-E interface. The ZFS file-system (bundled with the open source SmartOS operating system) is used to manage the data in a redundant configuration that allows any two disks in the 24 disk array to simultaneously fail without data loss. Due to the redundancy, each server has 60 TB of usable space that can be read at approximately 16 gigabits per second (Gbps). This high I/O bandwidth is critical for data backup, recovery and integrity verification.

### Distributed computing software setup

To fully utilize available CPU and I/O resources, we parallelized the data processing (*Dean and Ghemawat, 2008*). Thread-level parallelization inside a single process is the simplest approach and coordination between threads is orchestrated using memory shared between the threads. However, this approach only works for a single machine and does not scale to a cluster of computers. The typical approach to cluster-level parallelization is to coordinate the multiple parallel computations running both within a machine and across machines by exchanging messages between the concurrently running processes. The 'map-reduce' abstraction conceptualizes a computation as having two phases: a 'map' phase which first processes small subsets of the data in parallel and a 'reduce' phase which then serially aggregates the results of the 'map' phase. Since much of our data analysis is I/O limited (like computing statistics of spike waveform amplitudes), we developed a custom distributed computing infrastructure for 'map-reduce' like computations for the .NET platform. The major novelty in our framework is that rather than moving the output of the map computation over the network to a central node for performing the reduce computations, it instead moves the reduce computation to the machines containing the output of the map computations in the correct serial order. If the output of the map computation is voluminous compared to the output of each step of the reduce computation then our approach consumes significantly less time and I/O bandwidth. We have used this framework both in a virtual cluster of 5 virtual machines running on each of the afore-mentioned SmartOS-based storage servers and in Microsoft's commercial cloud computing platform – Windows Azure.

## Analysis of neural data

### Automated spike-sorting algorithm

We developed a new and automated spike-sorting algorithm (*Poddar et al., 2017*) that is able to efficiently track single units over long periods of time (code available at https://github.com/Olveczky-Lab/FAST; copy archived at https://github.com/elifesciences-publications/FAST). Our offline algorithm first identifies the spikes from the raw recordings and then performs two fully automated processing steps. A list of important algorithm parameters and their recommended values is provided in *Table 1*.

### Spike identification

We first extract spike snippets, that is, the extracellular voltage traces associated with action potentials, automatically from the raw data. A schematic of this process is shown in *Figure 2—figure supplement 1*. First, the signal from each channel $s_{ch}(t)$ is partitioned into 15 s blocks with an additional 100 ms tacked onto each end of the block to account for edge effects in filtering. Then, for each block of each channel, the raw data is filtered with a fourth order elliptical band-pass filter (cut-off frequencies 300 Hz and 7500 Hz) first in the forward then the reverse direction to preserve accurate spike times and spike waveforms. For each sample in each block, the median across all channels is subtracted from every channel to eliminate common mode noise that may arise from rapid movement or behaviors such as eating (*Figure 2—figure supplement 4*). Finally, a threshold crossing algorithm is used to detect spikes independently for each tetrode (*Figure 2—figure supplement 1*). In our recordings, we use a threshold of 50 μV, which corresponds to about 7 times the median absolute deviation (m.a.d.) of the signal (average m.a.d. = 7.6 ± 0.6 μV in our recordings), but we

**Table 1.** List of parameters for the fully automated steps of FAST and their recommended values.

| Step | Parameter | Details | Recommended value |
|---|---|---|---|
| Snippeting | block size | Length of data segment to filter and snippet per block | 15 s |
| Snippeting | end padding | To account for edge effects in filtering | 100 ms |
| Snippeting | waveform samples pre and post | Samples per spike snippet to record before and after peak (not including peak sample) | 31 and 32 samples (total 64 samples @ 30 kHz sampling) |
| Snippeting | spike detection threshold | Threshold voltage to detect spikes | 7 × median absolute deviation of recording |
| Snippeting | spike return threshold | Threshold voltage below which the spike detection process is reset | 3 × median absolute deviation of recording |
| Snippeting | spike return number of samples | Number of samples the recorded voltage must stay under the spike return threshold to reset the spike detection process | 8 samples (0.27 ms @ 30 kHZ sampling rate) |
| Clustering: Step 1 | spikes per block | Number of spikes per block for local clustering | 1000 spikes |
| Clustering: Step 1 | minimum and maximum SPC temps | Range of temperatures for super-paramagnetic clustering (SPC) | 0, 15 |
| Clustering: Step 1 | Threshold for recursive merging of SPC tree ('a') | Threshold distance between leaf node and its parent node in the SPC tree below which the leaf node is merged with its parent. | 20 μV |
| Clustering: Step 1 | Minimum cluster size | Min. number of spikes in cluster to compute cluster centroid | 15 spikes |
| Clustering: Step 1 | Number of rounds of local clustering | Rounds of iterative multi-scale clustering | 4 |
| Clustering: Step 2 | Spike centroids per block | Number of spike cluster centroids per block for generating SPC tree prior to segmentation fusion | 1000 centroids |
| Clustering: Step 2 | Cluster trees per block | Number of cluster trees per block for segmentation fusion | 10 cluster trees |
| Clustering: Step 2 | Segmentation fusion block overlap | Number of cluster trees overlap between blocks for segmentation fusion | 5 cluster trees |
| Clustering: Step 2 | 's', 'k' | Parameters for sigmoid scaling of link weights to be between 0 and 1 in segmentation fusion | s = 0.005, k = 0.03 |
| Clustering: Step 2 | Link similarity threshold | Similarity threshold for link weights between cluster trees in segmentation fusion | 0.02 (range 0–1) |
| Clustering: Step 2 | Link similarity threshold for straggler nodes | Similarity threshold for joining leftover nodes to an existing chain in segmentation fusion | 0.02 (range 0–1) |

DOI: https://doi.org/10.7554/eLife.27702.022

note that this value is not hard-coded in FAST and can be changed to an exact multiple of the measured noise. After detecting a threshold crossing, we find the sample that corresponds to the local maximum of the event, defined as the maximum of the absolute value across the four channels of a tetrode. A 2 ms (64 sample) snippet of the signal centered on the local maximum is extracted from all channels. Thus, each putative spike in a tetrode recording is characterized by the time of the local maximum and a 256 dimensional vector (64 samples x 4 channels).

Each 15 s block of each tetrode can be processed in parallel. However, since the number of spikes in any given 15 s block is not known in advance, the extracted spike snippets must be serially written to disk. Efficiently utilizing all the cores of the CPUs and simultaneously queuing disk read/write operations asynchronously is essential for keeping this step faster than real-time. In our storage server, the filtering/spike detection step runs 15 times faster than the acquisition rate. To extract LFPs, we down-sample the raw data 100-fold (from 30 kHz to 300 Hz) by two applications of a fourth order 5-fold decimating Chebychev filter followed by a single application of a fourth order 4-fold decimating Chebychev filter. After extracting the spike snippets and the LFPs, the raw data is deleted, resulting in a 5–10 fold reduction in storage space requirements.

A typical week-long recording from 16 tetrodes results in over a billion putative spikes. While most putative spikes are low amplitude and inseparable from noise, the spike detection threshold cannot be substantially increased without losing many cleanly isolatable single units. Assigning many

billion putative spikes to clusters corresponding to single units in a largely automated fashion is critical for realizing the full potential of continuous 24/7 neural recordings.

## Automatic processing step 1. local clustering and de-noising

This step of the algorithm converts the sequence of all spike waveforms $\{x_i\}_{i=1}^N$ from a tetrode to a sequence of averages of spike waveforms $\{y_i\}_{i=1}^M$ with each averaging done over a set of approximately 100 raw spike waveforms ($100M \cong N$) with very similar shapes that are highly likely to be from the same unit. The output of this step of the algorithm is a partitioning of $N$ spike waveforms into $M$ groups. $y_i = \frac{1}{N_i} \sum_{j=1}^{N_i} x_{i_j}$, $\sum_{i=1}^M N_i = N$. Local clustering followed by averaging produces de-noised estimates of the spike waveform for each unit at each point in time. This results in a dataset of averaged spike waveforms that is about a hundred times smaller than the original dataset greatly aiding in speedily running the remaining half of the spike sorting algorithm and in visualizing the dataset.

## Super-paramagnetic clustering for dataset compression

Clustering is inherently a scale-dependent problem, that is, the 'right' number of clusters in a given dataset depends on the scale being considered. At the coarsest scale, all points can be considered members of a single cluster and at a very fine scale each point belongs to its own cluster. A formal, mathematically precise way of characterizing this tradeoff is to think of clustering as lossy compression (*Slonim et al., 2005*). The amount of loss is defined as the amount of information lost by replacing each point in a cluster with their 'average' and the compression comes from characterizing the entire dataset with just the cluster averages. A simple loss measure is the sum of squared distances between each point in a cluster and the cluster centroid, that is, the within cluster variance. If each point is assigned its own unique cluster then the loss would be zero. Conversely, if all points were assigned the same cluster then the loss would simply be the variance of the entire set of points. For intermediate amount of loss between these two extremes, the fewest number of clusters, that is, the largest amount of compression, with at most that much loss, can, in principle, be computed. Conversely for a given number of clusters, one can compute the clustering that results in the smallest amount of loss.

We use the super-paramagnetic clustering (SPC) algorithm (*Blatt et al., 1996*) in our spike sorting pipeline partly because it parameterizes the loss-compression tradeoff discussed above with a 'temperature' parameter. At low temperatures, the algorithm assigns all points to a single cluster. As the temperature parameter is increased new clusters appear until, at very high temperatures, each point is assigned its own cluster. Units with large spike waveforms or very distinctive spike shapes appear at relatively low temperatures. However, other units often appear at relatively high temperatures and clusters at higher temperatures often do not include spikes in the periphery of the cluster. In existing uses of this algorithm for spike sorting the 'right' temperature for each cluster is selected manually (*Quiroga et al., 2004*). Often several clusters at a higher temperature need to be manually merged as they all correspond to the same single unit.

The SPC algorithm also requires a distance measure between pairs of points. In previous approaches to spike sorting, a small number of features (on the order of 10) are extracted from the full 256 dimensional dataset and the Euclidean distance between points in this new feature space is used as the distance measure for clustering (*Quiroga et al., 2004*). In our experience the number of coefficients that are necessary to adequately capture the distinction between similar, but well isolated units varies substantially depending on the number of units being recorded on a tetrode and the signal-to-noise ratio of the recording. We find that simply using the Euclidean distance in the raw 256 dimensional space avoids this problem without being computationally prohibitive.

## Iterative multi-scale clustering for dataset compression

Two considerations determine the size of the batch for local clustering. First, SPC requires computing distances between every pair of points, making the algorithm quadratic in the number of points being clustered in one batch. In a typical desktop-class PC, a window size of 10000 spikes runs 8 times slower than real-time, whereas 1000 spike batches run faster than real-time. Second, changes in the spike waveform of a unit over time means that the space occupied by points belonging to a single cluster increases with the size of the temporal window (i.e. batch size), which in turn decreases

the separation between clusters. Both of these considerations favor clustering relatively fewer spikes in a batch, and our algorithm does it in batches of 1000 spikes.

However, different neuron types can have very different firing rates (*Figure 5*). Therefore, a 1000 spike window might contain just a few or even no spikes from low firing rate units. To overcome this problem, we developed a multi-resolution approach for the local clustering step. It identifies clusters corresponding to units with high firing rates, then removes them from the dataset and then re-clusters the remaining spikes, repeating this process iteratively (*Figure 2A–C* and *Figure 2—figure supplement 2A–D*).

## Detailed algorithm for local clustering and de-noising

The detailed steps of the algorithm for multi-resolution local clustering are described below and a schematic of the whole process is presented in *Figure 2—figure supplement 2A–D*.

1. The set of all spike waveforms is partitioned into blocks of 1000 consecutive points. Therefore, the first block contains points $\{x_1, \ldots, x_{1000}\}$, the second block contains $\{x_{1001}, \ldots, x_{2000}\}$ and so on.

2. An SPC cluster tree is generated for each block independently. This is computed by first clustering the set of 1000 points at a range of temperatures $T_i = 0.01i$, $0 \leq i \leq 15$. This process assigns a cluster label to each point at each temperature. This matrix of cluster labels is then converted to a tree where each node in the tree corresponds to a cluster at some temperature, i.e. a subset of the 1000 points. The root node of the tree (depth 0) corresponds to a single cluster containing all 1000 points. The children of the root node (depth 1 nodes) correspond to a partition of the set of 1000 points based on the cluster labels at temperature 0.01. For each node in depth 1, the children of that node (depth 2 nodes) correspond to a partition of the points associated with that node based on the cluster labels of those points at A Series of Monographs and Textbooks (Academic Press). temperature 0.02. This is repeated for all temperatures to construct the full tree with depth equal to the number of temperature increments.

3. Each cluster tree is collapsed into a partition (a clustering) of the set of 1000 points (*Figure 2—figure supplement 2B*). The simplest technique for getting a partition from an SPC cluster tree is to use all the nodes at a fixed depth, i.e. clustering at a fixed temperature. In practice, this approach suffers from major drawbacks. The lowest temperature at which a cluster first separates from its neighbors varies from unit-to-unit and depends on the signal-to-noise ratio of the spike waveform, how distinct the spike waveform of that unit is, etc. Also, when units appear at relatively high temperatures, the clusters corresponding to single units at those temperatures don't include many spikes at the periphery of those clusters. Therefore, instead of using a single temperature we recursively merge leaves of the cluster tree based on the loss-compression tradeoff discussed above to generate a partition. This is done by recursively collapsing the cluster tree one level at a time, a form of agglomerative hierarchical clustering (*Anderberg, 1973*). Specifically,

   a. For each leaf node in the cluster tree, the similarity between the leaf node and its parent is first calculated. Let $L = \{i_1, \ldots, i_N\}$ be the leaf node which is specified by the indices of the subset of the 1000 points belonging to that node. Similarly, let $P = \{j_1, \ldots, j_M\}$ be the parent of the leaf node. Note that $L \subseteq P$. Let $l = \frac{1}{N}\sum_{k=1}^{N} x_{i_k}$ be the average spike waveform of the leaf node and $p$ be the average spike waveform of its parent. Let $d_L = \sum_{k=1}^{N} \|x_{i_k} - l\|^2$ be the total distance of points in the leaf node from their average and $d_P = \sum_{k=1}^{N} \|x_{i_k} - p\|^2$ be the distance from the parent node. The difference $\frac{\sqrt{d_P - d_L}}{\nu} = a_L$, where $\nu$ is the degrees of freedom (256 waveform samples for a tetrode recording), measures how well the parent node approximates the leaf node, that is, how much additional loss is incurred in approximating the points in the leaf node with the cluster corresponding to the parent node.

   b. Let $L = \{L_i\}$ be the set of all N leaf nodes sharing the same parent node $P$. The set of leaf nodes that are poorly approximated by their parent (the well-isolated nodes $I$) are considered distinct clusters. $I \subseteq L$, where $L_i \in I$ if $a_{L_i} > a$. This encodes the intuition that if a cluster at a given temperature splits into multiple sub-clusters at the next higher

temperature that are each quite similar to the parent cluster, then treating each of these sub-clusters as distinct clusters is inappropriate. The parameter $a$ provides an intuitive tradeoff between missing distinct clusters that appear at high temperatures and classifying spikes in the periphery of a single cluster into multiple distinct clusters. Let $M$ be the number of elements in $I$. If any of the remaining $N - M$ nodes are well approximated by one of the well-isolated nodes then they are merged together. For $L_i \in L/I$, if $L_j \in I \min d_{L_j} - d_{L_i} \< a$, i.e. if node $L_j$ approximates node $L_i$ well then they are merged. Merging a set of nodes corresponds to creating a node containing all the points from each of the nodes being merged. This yields a set of augmented well-isolated nodes. Any remaining nodes, i.e. non-well-isolated nodes that are also not well-approximated by any of the well isolated nodes, are merged with each other. Therefore, this step results in converting the set of N leaf nodes sharing a parent into a set of $M$ or $M + 1$ nodes formed by merging some of them together. We set $a$ to 20 μV, a value that was empirically determined to favor over-clustering of the SPC tree so as to prevent the incorporation of spikes from distinct units into the same local cluster.

 c. A depth $D$ tree is converted into a depth $D - 1$ tree by replacing all the leaf nodes and their parents with the merged nodes derived in the previous step.

 d. Step a – c are repeated recursively until the tree is of depth 1. The leaf nodes of this tree which are typically vastly fewer in number than the total number of leaf nodes of the original tree correspond to a partition of the set of 1000 points of each block.

4. The centroid of each cluster from the previous step containing at least 15 points contributes one element to the output of the local clustering step, the set of averaged spike waveforms $\{y_i\}$ (*Figure 2D* and *Figure 2—figure supplement 2C*). In our datasets, clusters with at least 15 spikes in a 1000 spike window correspond to firing rates of approximately 1 Hz or greater, on average. The points belonging to the remaining clusters, that is, ones with fewer than 15 points, are all pooled together, ordered by their spike time and become the new $\{x_i\}$. The number of spikes in this new subset is approximately 10% of the original. Steps 1–4 are used to locally cluster this new subset of spikes and produce a second set of averaged spike waveforms $\{y_i\}$. This process of re-clustering the low-density clusters is repeated two more times. The averaged spike waveforms from all four scales are then grouped together to form the full set $\{y_i\}_{i=1}^M$ and ordered by the median spike time of set of spikes that were averaged to generate each $y_i$. This process results in an assignment of over 98% of the original set of $N$ spikes to a cluster with at least 15 spikes in one of the 4 scales. The firing rate of units in clusters with at least 15 spikes at the fourth scale is about 0.01 Hz in the sample dataset.

## Automatic processing step 2. sorting and tracking de-noised spike waveforms

This is FAST's principal single-unit clustering and chaining step, that takes the sequence of averaged spike waveforms $\{y_i\}_{i=1}^M$ computed in the previous step and identifies the subsets of $\{y_i\}$ that correspond to the same putative single unit. A subset of $\{y_i\}$ is considered to be the same single unit if distances between $y_i$ and $y_{i+1}$ are sufficiently small for the entire subset. As in the previous step, the set $\{y_i\}_{i=1}^M$ is first partitioned into blocks of 1000 consecutive points and an SPC cluster tree is independently computed for each block, this time for the temperature range $T_i = 0.01i, \ 0 \le i \le 10$. The rationale for the second round of SPC is to pool together averaged spike waveforms corresponding to the same unit both within and across the four iterations of local clustering and de-noising in Step 1 of FAST. In our datasets, Step 1 of FAST yields, on average, 8.0 ± 2.6 (mean ±SD, n = 80 tetrodes) cluster centroids per 1000 spike block. This means that a 1000 centroid block effectively encompasses ~125 blocks in Step 1, corresponding to 125000 spikes. Therefore, our multi-scale clustering approach ensures that single unit clusters are actually defined over relatively large windows spanning >100000 spikes, thereby resulting in accurate clustering.

 Next, we use a binary linear programming algorithm inspired by a computer vision problem called segmentation fusion (*Vazquez-Reina et al., 2011*) to identify the nodes in each cluster tree that correspond to the same single unit.

## Binary linear programming allows units to be tracked over time

Tracking multiple single units over time from a sequence of cluster trees requires first selecting a subset of the nodes of each cluster tree that correspond to distinct units, followed by matching

nodes from adjacent cluster trees that correspond to the same unit. Doing these steps manually is infeasible because of the large volumes of data. In our datasets, the local-clustering step results in ~100 million averaged spikes from the original set of ~10 billion spikes per rat. This produces a set of ~100,000 cluster trees making manual tracking impossibly labor intensive. We instead adapted the segmentation fusion algorithm, invented to solve the problem of reconstructing the full 3D structure of axons, dendrites and soma present in a volume of neuropil from a stack of 2D electron microscopy sections (*Kaynig et al., 2015*; *Vazquez-Reina et al., 2011*). A cluster tree is analogous to a multi-resolution segmentation of the 2D image. Identifying nodes across cluster trees that correspond to the same single unit is thus analogous to identifying the same segment across 2D sections as a neurite courses through the neuropil.

The algorithm finds a set of sequences of nodes from the sequence of cluster trees, where each sequence of nodes corresponds to a well isolated single unit. This is done by first enumerating all the nodes in all the cluster trees and all possible links between nodes in adjacent cluster trees. Then a constrained optimization problem is solved to find a subset of nodes and links that maximize a score that depends on the similarity between nodes represented by a link and the 'quality' of a node. This maximization is constrained to disallow assigning the same node to multiple single units and to ensure that if a link is selected in the final solution then so are the nodes on each side of the link.

## Details of the binary linear programming algorithm

Each step of the binary linear programming algorithm is detailed below (see also *Figure 2D,E* and *Figure 2—figure supplement 2E,F*).

1. The sequence of cluster trees is grouped into blocks of 10 consecutive trees with an overlap of 5 cluster trees. Solving the binary linear program for blocks larger than 10 trees is computationally prohibitive.

2. The segmentation fusion algorithm is run independently for each block of 10 cluster trees (*Figure 2—figure supplement 2E*). Let $\left\{ \left\{ C_j^i \right\}_{j=1}^{N_i} \right\}_{i=1}^{10}$ be the set of binary indicator variables representing all nodes in all 10 cluster trees where the cluster tree indexed by $i$ contains a total of $N_i$ nodes. The total number of nodes is $N = \sum_{i=1}^{10} N_i$. Let $\left\{ \left\{ L_{jk}^i \right\}_{j,k=1,1}^{N_i,\ N_{i+1},} \right\}_{i=1}^{9}$ be the variables representing the set of all links between adjacent cluster trees. Link $L_{jk}^i$ connects clusters $C_j^i$ and $C_k^{i+1}$. The total number of links is $M = \sum_{i=1}^{9} N_i N_{i+1}$. Solving the linear program requires choosing a $\{0,1\}$ value for each of the $N + M$ binary variables that maximizes the objective function $ij\sum C_j^i \theta_j^i + \sum_{ijk} L_{jk}^i \left( \theta_{jk}^i - 0.02 \right)$. The objective function is a weighted linear sum of all the binary variables where the cluster weights $\theta_j^i$ represent the 'quality' of the cluster and the link weights $\theta_{jk}^i$ represent the similarity of the clusters joined by the link. The link weights are numbers in the range $(0,1)$. The threshold of 0.02 serves to give negative weight to links between sufficiently dissimilar clusters, effectively constraining the value of the variables representing those links to 0. This objective function is to be optimized subject to three sets of constraints. The first, $\sum_k L_{jk}^i \leq C_j^i$, enforces the constraint that if the node variable $C_j^i$ is assigned a value of 1 then out of all the outgoing links from the node $\left\{ L_{jk}^i \right\}_{k=1}^{N_{i+1}}$, at most one is chosen (*Figure 2—figure supplement 2F*, top). Similarly, the second set of constraints, $\sum_j L_{jk}^i \leq C_k^{i+1}$, enforces the requirement that at most one incoming link to a node is chosen (*Figure 2—figure supplement 2F*, middle). The third set of constraints enforces the requirement that for each of the 1000 points in a cluster tree at most one of the nodes containing that point is chosen (*Figure 2—figure supplement 2F*, bottom). This translates to inequalities $\sum_{k \in I_j} C_k^i \leq 1$ where the set of indices $I_j$ represents nodes in the path from the root of cluster tree $i$ to the $j^{th}$ leaf node of the cluster tree. Therefore, the total number of constraints of this type for each cluster tree is the number of leaf nodes in that cluster tree. The link weight $\theta_{jk}^i$ is the Euclidean distance between the average spike waveform of clusters $C_j^i$ and $C_k^{i+1}$ non-linearly scaled by a sigmoid function to fall in the range $(0,1)$. If $d$ is the distance

then $\theta = \frac{a}{1+a}$, $a = \exp\left(\frac{-(d-k)}{s}\right)$, $s = 0.005$, $k = 0.03$. The parameter $s$ controls the steepness of the sigmoid and the parameter $k$ sets the distance $d$ at which $\theta = 0.5$. The cluster weight $\theta_j^i$ gives preference to clean well-isolated clusters, i.e. clusters that appear at low temperatures and retain most of their points across a large temperature range. Let $N^{(0)}$ be the number of points in the cluster corresponding to $C_j^i$. Let $C_k^i$ be the largest cluster amongst the child nodes of $C_j^i$ and let $N^{(1)}$ be the number of points in $C_k^i$. Similarly let $N^{(2)}$ be the number of points in the largest cluster among the child nodes of $C_k^i$. Given the sequence of cluster sizes $N^{(0)}, N^{(1)}, \ldots, N^{(a)}$ where $N^{(a)}$ is the number of points in a leaf node of cluster tree, $\theta_j^i$ is defined as $N^{(0)}/\left(N^{(0)} + \ldots + N^{(a)}\right)$. This measure of cluster quality penalizes clusters that split into smaller clusters at higher temperatures and clusters that only appear at high temperatures.

3. The results of the previous step, i.e. the subset of the M links of each block that maximizes the objective function, are finally combined to produce a sorting that tracks single units over long time periods despite gradually changing waveforms. Links that are part of two instances of the segmentation fusion procedure due to the overlap mentioned in step 1 are only included in the final solution if both linear programs include them. The set of links chosen by the segmentation fusion algorithm are chained together to get long chains of clusters. For instance if links $L_{jk}^i$, $L_{kl}^{i+1}$, $L_{lm}^{i+2}$ are assigned values of 1 in the solution to the segmentation fusion linear program then all points in clusters $C_j^i$, $C_k^{i+1}$, $C_{jl}^{i+2}$, $C_m^{i+3}$ belong to the same chain and hence the same single unit. Each point that does not belong to any chain is assigned to the chain containing points most similar to it (as measured using the sigmoidal distance of step 2) as long as the similarity $\theta > 0.02$ (again the same threshold as used in step 2).

## Merging the chain of nodes and links identified by the binary linear programming algorithm

Often, spike waveforms have multiple equal amplitude local extrema. Since the waveforms are aligned to the local extrema with the largest amplitude during the spike identification phase, different waveforms from the same unit can be aligned to different features of the waveform. This results in multiple chains for the same single unit since the Euclidean distance between waveforms aligned to different features is very large. This is remedied by merging chains that contain spikes from the same recording interval if the translation-invariant distance between the spike waveforms of the chains is sufficiently low in the overlap region. The translation invariant distance is computed by first calculating the distance between a pair of spike waveforms for a range of relative shifts between the pair and then taking the minimum of this set of distances. Overlapping chains with the smallest translation-invariant distance are first merged. This is done recursively until either no overlap between chains remains or overlapping chains have distinct spike waveforms and hence correspond to different simultaneously recorded single units.

## Drift correction rate of the FAST algorithm

The time-period over which FAST assumes that spike waveforms are stable (the 'tracking time-scale') is a function of both the mean event detection rate and the number of distinct units simultaneously recorded on the electrode. The event detection rate sets the time-scale of the 1000 spike local-clustering block. The number of simultaneously recorded units scales with the average number of clusters in each 1000 spike block, and sets the time-scale of the 1000 centroid block.

In our recordings, the mean event detection rate was 62.4 ± 39.0 Hz (mean ± SD, n = 80 tetrode recordings), resulting in an average duration of 16 s for a 1000 spike block. If all spikes belonged to the same unit and the local clustering step yielded only 1 centroid per 1000 spike block, a 1000 centroid block would span 16*1000/1 = 16000 s or 4.5 hr; this represents the slowest possible tracking rate. On the other hand, if we were recording from 1000/15 ∼ 66 distinct units with equal firing rates (so as to ensure that the number of spikes per unit per block exceeded the minimum cluster size of 15), the local clustering step would yield 66 centroids per 1000 spike block and a 1000 centroid block would span only 16*1000/66 = 242 s or 4 min; this represents the fastest possible tracking rate. In our own recordings, local clustering in Step 1 yielded centroids at a rate of 8.0 ± 2.6 per 1000 spike block. This yields an approximate tracking time-scale of 16*1000/8 = 2000 s or 33 min.

The 'tracking time-scale' should be treated more as a guideline than an upper limit on FAST's drift correction rate. This is because FAST employs superparamagnetic clustering which can also sort the non-Gaussian shaped clusters that are an outcome of electrode drift (*Quiroga et al., 2004*; *Rey et al., 2015b*).

## Visualization and manual verification

The final output of the unsupervised spike sorting algorithm consists of long chains (median length = 10.2 hr) corresponding to single-unit spikes linked over time. However, our algorithm does not link spike chains over discontinuities in recording time (i.e. across recording files), or in the case of rapid changes in spike shape that occasionally occur during the recording, or when spike amplitudes drift under 75 µV for brief periods. In such cases, we have to link, or 'merge' chains across these discontinuities.

In order to visualize, merge and manually inspect the unsupervised chains, we developed a MAT-LAB program (*Dhawale et al., 2017a*) with a graphical user interface (available at https://github.com/Olveczky-Lab/FAST-ChainViewer; copy archived at https://github.com/elifesciences-publications/FAST-ChainViewer). Our program allows users to semi-automatically merge chains belonging to the same unit across discontinuities, based on end-to-beginning similarity in their spike waveforms and inter-spike interval distributions (correlation coefficient between spike-waveforms or ISI distributions >= 0.9). We merge chains only if the time-gap between the end of one chain and the beginning of the next is less than 5 hr of recording time, or up-to 24 hr in the case of gaps in recording time (which in the experiments we report on were, on average, 2 hr). In our dataset, the number of manual mergers required per unit was, on average, 14.7 ± 30.6 (mean ± SD), which translated to an average of 1.3 manual chain mergers per tetrode per day.

No automated spike-sorting method is perfect. Occasionally, the completely unsupervised section of the algorithm (Steps 1 and 2) makes errors in spike-sorting by merging spikes from multiple units in the same chain. This typically happens when spike amplitudes drift close to background noise, or when spike waveforms corresponding to distinct units are very similar to each other in segments of the recording. In order to identify such missorting, we visually inspect raw spike waveforms of those chains whose amplitudes are below 100 µV or whose average waveforms have a greater than 0.9 correlation to multiple preceding or succeeding chains using the manual spike-sorting MClust software (MClust 4.3, A. D. Redish et al). In the event of mis-sorting (only 2% of all unit chains, or 0.027 chains per tetrode per day), we manually split these chains using MClust.

We manually merge pairs of units with identical spike shapes, but distinct spike amplitudes, if their spike-time cross-correlograms are highly asymmetric around 0 ms and show a distinct refractory period, as these distinct clusters most likely arose from the same bursting unit. We typically exclude clusters whose spike-waveforms are complex and are likely to result from simultaneous spiking from multiple units.

In our hands, it takes about 8 human hours to perform manual verification of the automated output of FAST applied to a 3 month long recording on a single tetrode, or approximately 130 hrs to process all 16 tetrodes in our recording array. Thus manual verification proceeds at a rate about 5.5 times faster than the rate of data-acquisition, which is nearly twice as fast as the slowest automated step of FAST (Step 1). A recommended workflow for the manual steps is illustrated in *Figure 2—figure supplement 3*.

All subsequent analysis of neural data was carried out using custom scripts in Matlab (Mathworks). See the sections below for detailed description of the analyses.

## Validation of FAST's spike sorting performance with ground-truth datasets

### Processing of simultaneous extracellular-intracellular recording datasets

We processed these datasets (*Henze et al., 2009*) as described in earlier studies (*Ekanadham et al., 2014*; *Harris et al., 2000*). Briefly, we high-pass filtered the intracellular trace at 250 Hz with a third order Butterworth filter and detected peaks larger than four standard deviations of the baseline noise to identify intracellular (ground truth) spike times. Since FAST requires a minimum of ~$10^6$ spikes for clustering, we concatenated 20–50 identical copies of the relatively short duration recordings from (*Harris et al., 2000*) to construct a 'longer' recording. After using FAST to spike-sort the

concatenated recording, we measured its median sorting performance over all 20–50 copies of each dataset.

## Generation of synthetic long-term extracellular recordings

We created realistic synthetic long-term extracellular recordings (sample available at *Dhawale et al., 2017b*) (n = 6, *Figure 3C*) as follows. To incorporate realistic signal-independent variability arising from thermal or amplifier noise, we identified a tetrode in our dataset with a negligible number of detected spikes. To this 'blank' recording, we added spikes from eight distinct simulated units. To do this, we first generated a 'spike library' of ~1000 different unit waveforms and firing rates varying over 3 orders of magnitude from our own long-term recordings in the motor cortex and striatum. From this library, we drew eight units at random per tetrode, ensuring that 20% of selected units were interneurons. For each unit, we generated spike times at 300 kHz precision over the 256 hr time-period of the simulated recording following a homogenous Poisson process with rate equal to the firing rate of that unit in our 'spike library', and an absolute refractory period of 1.5 ms. Spike waveforms corresponding to each unit were up-sampled to 300 kHz by cubic interpolation and scaled to an appropriate spike amplitude before being down-sampled to 30 kHz and added to the 'blank' tetrode recording. The up-sampling followed by down-sampling procedure was done to simulate jitter in the alignment of spike waveforms by the data acquisition and snippeting process (*Rossant et al., 2016*).

The spike amplitude of each unit drifted over time according to a geometric random walk with bounds, as shown in *equation 2* (*Rossant et al., 2016*). We also add independent signal-dependent variability that scales with the spike amplitude, as shown in *equation 2* (*Martinez et al., 2009*; *Rossant et al., 2016*).

$$A_{t+t} = A_t \exp\left(\sqrt{\beta t}\epsilon\right)(1 + \alpha\epsilon) \tag{2}$$

where $b_{min} \leq A_t \exp(\sqrt{\beta t}\epsilon) \leq b_{max}$ and $\epsilon \sim N(0,1)$

where $A_t$ - 6 is the amplitude of a unit's spike at time $t$ - 6, $A_t$ - 6 is the amplitude of unit's spike after time lag of $t$ - 6, $\beta t$ - 6 represents the variance of the geometric random walk, $\alpha$ - 6 represents the scaling parameter for the signal-dependent variability in spike amplitude, $\epsilon$ - 6 represents a random number drawn from the standard Gaussian distribution, and $b_{min}$ - 6 and $b_{max}$ - 6 represent the minimum and maximum bounds, respectively, on the random walk, for a given electrode. We empirically determined the values for $\alpha$ - 6 (0.1) and $\beta$ - 6 (10$^{-6}$ s$^{-1}$) by analyzing variations in spike amplitudes in our long-term recordings, as shown in *Figure 3—figure supplement 2A–B and C–D*, respectively. $b_{min}$ - 6 was set to 75 µV for all electrodes to ensure that spike amplitudes did not drift below the spike detection threshold, while $b_{max}$ - 6 was drawn independently for each electrode from an exponential distribution within the range 150 to 400 µV. The exponent of the $b_{max}$ - 6 distribution ($-0.005$ µV$^{-1}$) was empirically determined from the observed distribution of maximum spike amplitudes in our recording dataset (*Figure 3—figure supplement 2E*). Finally, we also added low amplitude, multi-unit 'background activity' to the recording representing the contribution of units farther away from the recording electrode (*Martinez et al., 2009*). This background activity consisted of 40 distinct units drawn from our spike library, each firing at 0.5 Hz to yield a total background firing rate of 20 Hz. Background spike amplitudes were drawn from a Gaussian distribution with mean 50 µV (our spike detection threshold) and standard deviation 25 µV.

## Calculation of best ellipsoidal error rate (BEER)

Our calculations of the BEER measure were similar to previous studies (*Ekanadham et al., 2014*; *Harris et al., 2000*). For each spike, we calculated a spike feature vector consisting of the top 3 principal component projections of the spike waveform on each of the 4 electrodes, yielding a total of 12 features per spike per tetrode. We then trained a support vector machine (SVM) classifier, using a training subset of the data, to find a quadratic boundary in feature space best separating spikes belonging to the unit of interest and all other detected spikes. We then tested the classification performance of the trained SVM on a separate test subset of spikes. As the synthetic datasets are extremely large - up to 5e7 spikes per tetrode - we used smaller subset of the spikes for training and testing the SVM (~1e5 spikes in each set), after verifying that this did not affect the accuracy of

the classifier (data not shown). To vary the relative importance of false positive versus false negative errors, we systematically varied their relative classification costs for the SVM over two orders of magnitude to yield receiver-operator characteristic (ROC) curves, as shown in *Figure 3—figure supplement 1A*. We considered the BEER error rate to be the minimum error on this ROC curve (i.e. assuming equal weight for false-positive and false-negative errors).

## Analysis of long-term recordings

### Cluster quality

After spike sorting, we computed standard measures of cluster quality, specifically the isolation distance (*Harris et al., 2000*), L-ratio (*Schmitzer-Torbert et al., 2005*) and fraction of inter-spike intervals violating the absolute refractory period (2 ms), for every unit within consecutive one-hour blocks. Isolation distances and L-ratios were calculated with four features for every electrode channel – the waveform peak, waveform energy ($L^2$ norm), and the projections of the spike waveforms on the first two principal components of all spike waveforms in the one-hour block. Only units meeting all of our quality criteria (isolation distance => 25, L-ratio <= 0.3, fraction ISI less than 2 ms <= 0.01) in a particular block were subjected to further analysis. The average signal-to-noise ratio (SNR) was for each unit was calculated by dividing the average spike amplitude by a consistent estimator of the standard deviation of the baseline – namely the median absolute deviation of the filtered recording multiplied by 1.4826. For every unit, we report its maximum SNR across the four electrode channels of a tetrode.

### Discontinuous tracking of units across days

We identified units present (>=50 spikes fired) in daily one-hour 'sessions' from 10 to 11 am, during execution of a skilled motor sequence task (*Kawai et al., 2015*). We computed four measures of unit similarity across consecutive days' sessions (*Figure 4—figure supplement 1A*), namely the Euclidean distance and correlation coefficient (*Emondi et al., 2004*) between average unit spike waveforms (waveforms recorded on different channels of a tetrode were concatenated together to yield a 256-element vector), the correlation coefficient between units' inter-spike interval histograms (*Dickey et al., 2009*) (see below), and the absolute difference in firing rates (*Fraser and Schwartz, 2012*). Correlation coefficients were Fisher transformed, and spike waveform Euclidean distance and firing rates were log-transformed to appear more Gaussian (*Dickey et al., 2009*; *Fraser and Schwartz, 2012*) (*Figure 4—figure supplement 1A*). Similarity metrics were calculated for all pairs of units recorded on the same tetrode across consecutive sessions/days and classified as either the 'same' or 'different' as per our FAST algorithm. Linear discriminant analysis (LDA) (*Tolias et al., 2007*) was then used to identify a similarity threshold that would best separate 'same' from 'different' unit pairs at false positive error rates ranging from 25% (*Dickey et al., 2009*) to 5% (*Fraser and Schwartz, 2012*), 1% and 0.2%. Here, false positive rates are defined as the proportion of 'different' units misclassified by the classifier as the 'same'. We also tried a quadratic classifier (*Dickey et al., 2009*; *Fraser and Schwartz, 2012*) but found no significant difference in performance compared to the linear classifier. LDA fits multidimensional Gaussians to the similarity distributions identified by 'same' or 'different' class labels. Therefore, assuming a uniform prior, we can calculate the posterior probability of a unit pair being part of the 'same' distribution and use it as a similarity score. Units were matched across consecutive daily 'sessions' if their similarity scores exceeded the threshold set by the expected false positive rate, with highest similarity unit pairs being matched first (*Emondi et al., 2004*). Thus, in order to track a unit across 20 days, it would have to be serially matched across 19 pairs of consecutive recording sessions.

### Identification of putative cell types

We used mean spike waveforms and average firing rate (*Barthó et al., 2004*; *Berke et al., 2004*) to separate units into putative cell types: medium spiny neurons (MSNs) and fast spiking neurons (FS) in the striatum, and regular spiking (RS) and fast spiking (FS) neurons in the cortex. Briefly, we performed k-means clustering of units based on their peak-normalized spike-waveform, concatenated with their log-transformed firing rates. The number of clusters (k = 2) was chosen by visual inspection of unit waveforms in three dimensional space, i.e. the first and second principal components of the spike waveform, and unit firing rate.

## Firing rate stability

Firing rate similarity compared across different days $i$ and $j$ for the same unit, or when comparing distinct units $i$ and $j$ recorded on the same day, was measured by the following formula:

$$FR\ similarity_{i,j} = 1 - 2\left(\frac{abs(FR_i - FR_j)}{FR_i + FR_j}\right) \tag{3}$$

where $FR$ is the firing rate. A firing rate similarity score of 1 means that the two firing rates $FR_i$ and $FR_j$ are identical while a firing rate similarity score of $-1$ implies maximum dissimilarity in firing rates, such that one of the firing rates is 0. When comparing firing rates for the *same unit across time*, we calculated average firing rate similarity scores for time-lags ranging from 1 to 20 days. When comparing firing rates *across units* and within the same time-bin, we averaged together all similarity scores for pairs of units of the same putative unit-type recorded in that time-bin.

## Stability of inter-spike interval (ISI) distribution

We computed ISI histograms using 50 log-spaced time-bins spanning inter-spike intervals from 1 ms to 1000 s. We used Pearson's correlation coefficient to measure the similarity between ISI distributions on different days for the same unit, and between ISI distributions of distinct units recorded on the same day. For each unit, as with the firing rate similarity measure described above, we averaged the ISI correlations for each time-lag ranging from 1 to 20 days. When comparing across units, we only compared their ISI distributions to other simultaneously recorded units of the same putative unit-type.

## Stability of state-tuning

We used the Pearson's correlation coefficient to estimate the similarity between units' state-dependent firing rate profiles (i.e. their state-tuning), either across days, or between different units recorded simultaneously.

## Spike-triggered averages (STA)

The spike-triggered average (STA) is commonly used in sensory neuroscience to characterize a neuron's 'receptive field' in stimulus space (*Marmarelis and Naka, 1972*; *Meister et al., 1994*). We adapted this method to estimate 'response fields' in movement space (*Cheney and Fetz, 1984*; *Cullen et al., 1993*; *Serruya et al., 2002*; *Wessberg et al., 2000*). For each unit, we computed spike-triggered averages of accelerometer power in three different behavioral states – eating, grooming and active exploration. We calculated the norm of the accelerometer signal as per *equation (1)*. The accelerometer norm was band-pass filtered between 0.5 Hz and 300 Hz using a third order Butterworth filter and zero-phase filtering. Spike-triggered averages (STA) were computed for each unit in a particular behavioral state by averaging, over spikes, the accelerometer signal norm in a window of ±500 ms centered on each spike. Formally,

$$STA_{acc}(\tau) = \frac{1}{n}\sum_{i=1}^{n}\|acc(t_i + \tau)\| \tag{4}$$

where $\tau$ is the time-lag, sampled at 300 Hz resolution, between spike and accelerometer power ranging from $-500$ to $+500$ ms, and $t_i$ the arrival times of unit spikes (a total of $n$ spikes) in a particular behavioral state. We only considered 'trigger-spikes' that were at least 500 ms away from a behavioral state-transition.

To measure the significance of the resultant STA kernels, we computed noise STAs after jittering spike times by ±500 ms. We used the standard deviation of the noise STAs across time-bins to calculate the Z-scored STAs. Any STA having at least 2 bins equal to or exceeding a Z-score of 5 was considered statistically significant and used for further analysis.

Just as with ISI distribution similarity, we calculated the similarity between STAs for the same unit on different days using Pearson's correlation coefficient. We also compared similarity of STAs across simultaneously recorded units, but only within the same behavioral state.

## Peri-event time histograms (PETHs)

We computed peri-event time histograms (PETHs) of instantaneous firing rates aligned to specific behavioral events during execution of a lever-pressing task (*Kawai et al., 2015*). We computed PETHs in windows ± 200 ms around the first lever-press event of a trial, as well as at times of nose-pokes into the reward port following successful trials. In order to restrict our analysis of neural activity to periods of stereotyped behavior, we selected only rewarded trials that followed previously rewarded trials (to control for the rat's starting position), and these trials' inter-press intervals had to be within 20% of the target inter-press interval of 700 ms (to ensure movement stereotypy). To estimate a unit's instantaneous firing rate on each trial, we convolved its event-aligned spike train with a Gaussian kernel ($\sigma$ = 20 ms). These firing rates were then averaged over the selected trials from a given day to yield the unit PETH. To determine whether a particular PETH had statistically significant modulation in firing rate, we estimated the p-value and Z-score for each bin in the PETH using a bootstrap approach. P-values were then pooled across days for all lever-press or nose-poke PETHs for a particular unit using Fisher's method. A unit was deemed to have significant modulation in its time-averaged PETH if at least 2 bins had p-values<1e-5. At this significance threshold the probability of getting >1 false positive over 1000 neurons and 20 time-bins is less than 0.01.

Similarity across PETHs for the same unit across days, or between different units on the same day was computed using the Pearson's correlation coefficient, similar to the measurement of similarity between ISI distributions and STAs. Similarities in lever-press and nose-poke PETH similarities were averaged for each unit for a particular time-lag (ranging from 1 to 20 days).

## Cross-correlograms

We computed cross-correlograms for all unit pairs recorded simultaneously using spikes recorded in all labeled behavioral states (*Figure 6*) or in specific behavioral states (*Figure 7D,E*). For each unit pair, we only calculated correlograms on days on which both units were simultaneosly recorded for at least 60% of the time (i.e., >14 of 24 hr). Correlograms were calculated in 1 ms bins and smoothed with a 3 ms boxcar filter. Correlograms for unit pairs recorded on the same tetrode are, by design, zero at zero lag since only one spike event is allowed per tetrode in a 1 ms bin. For these pairs, the average of the preceding and following bins was used to approximate the value of the correlogram at zero lag. Correlations were considered significant if two consecutive bins of the correlogram were above a threshold set at three standard deviations of a shuffled correlogram computed for the same pair (one spike train was shuffled by adding ±400 ms random jitter to each spike). Only correlograms that had at least 5000 spikes could be considered significant. The normalized amplitude of each correlogram was measured as its maximum peak (positive correlation) or trough (negative correlation), averaged over 3 bins around the peak/trough and normalized by the average bin height of the shuffled correlogram. Thus an amplitude >1 indicates a positive correlation whereas an amplitude <1 indicates negative correlations. The lag was defined as the bin location of the peak/trough in the correlogram.

## Stability of correlograms

We used the Pearson's correlation coefficient to quantify the similarity between correlogram distributions on different days.

## Tests of statistical significance

We used the Kruskal-Wallis one-way analysis of variance test to determine whether differences in within-unit similarity measures of firing rate, ISI distribution, interneuronal correlograms, behavioral state modulation of firing rate, spike-triggered averages and PETHs, over time-lags of 1 to 20 days were significant when compared to each other or to similarity measures computed across units. P-values were corrected for multiple comparisons by the Tukey-Kramer method. We used an alpha value of 0.05.

## Acknowledgements

We thank Michelle Choi for behavioral scoring of videos, Mahmood M. Shad for help benchmarking the speed of the FAST algorithm, and Aliaksandr Iuleu for help with figures.

## Additional information

### Funding

| Funder | Grant reference number | Author |
|---|---|---|
| Star Family Challenge Award | | Bence P Ölveczky |
| Human Frontier Science Program | | Steffen BE Wolff |
| Life Sciences Research Foundation | | Ashesh K Dhawale |
| Charles A. King Trust | | Ashesh K Dhawale |
| National Institute of Neurological Disorders and Stroke | R01 NS099323-02 | Bence P Ölveczky |

The funders had no role in study design, data collection and interpretation, or the decision to submit the work for publication.

### Author contributions
Ashesh K Dhawale, Conceptualization, Data curation, Software, Formal analysis, Validation, Investigation, Methodology, Writing—original draft, Writing—review and editing; Rajesh Poddar, Conceptualization, Data curation, Software, Formal analysis, Investigation; Steffen BE Wolff, Data curation, Formal analysis; Valentin A Normand, Software, Formal analysis, Visualization; Evi Kopelowitz, Software, Formal analysis; Bence P Ölveczky, Conceptualization, Supervision, Funding acquisition, Investigation, Methodology, Writing—original draft, Project administration, Writing—review and editing

### Author ORCIDs
Ashesh K Dhawale [iD] https://orcid.org/0000-0001-7438-1115
Bence P Ölveczky [iD] http://orcid.org/0000-0003-2499-2705

### Ethics
Animal experimentation: This study was performed in strict accordance with the recommendations in the Guide for the Care and Use of Laboratory Animals of the National Institutes of Health. All of the animals were handled according to approved institutional animal care and use committee (IACUC) protocols (#29-15) of the Harvard University. All surgery was performed under isoflurane anesthesia, and every effort was made to minimize suffering.

### Decision letter and Author response
Decision letter https://doi.org/10.7554/eLife.27702.026
Author response https://doi.org/10.7554/eLife.27702.027

## Additional files

### Supplementary files
• Transparent reporting form
DOI: https://doi.org/10.7554/eLife.27702.023

### Major datasets
The following dataset was generated:

| Author(s) | Year | Dataset title | Dataset URL | Database, license, and accessibility information |
|---|---|---|---|---|
| Mokri Y, Salazar RF, Goodell B, Baker J, Gray CM, Yen SC | 2017 | Synthetic tetrode recording dataset with spike-waveform drift | https://doi.org/10.5281/zenodo.886516 | Publicly available at the OpenAIRE project via Zenodo (https://zenodo.org/) |

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
