## [Decision Letter]

Thank you for submitting your article "Automated long-term recording and analysis of neural activity in behaving animals" for consideration by *eLife*. Your article has been reviewed by three peer reviewers, and the evaluation has been overseen by Andrew King as the Senior Editor and Reviewing Editor. The following individuals involved in review of your submission have agreed to reveal their identity: Frédéric Theunissen (Reviewer #1); Karunesh Ganguly (Reviewer #2); Nicholas Steinmetz (Reviewer #3).

The reviewers have discussed the reviews with one another and the Reviewing Editor has drafted this decision to help you prepare a revised submission.

Summary:

This manuscript describes an experimental set up and, more significantly, an automated spike sorting algorithm that allows the activity of ensembles of neurons to be tracked over multiple recordings days in behaving rats. Using simulations and analyses of real data, the authors show that this new approach (which they call Fully Automated Spike Tracker or FAST) outperforms other state-of-the-art spike sorting methods and is able to cope with movement of the brain relative to the electrodes. The recordings from motor cortex and striatum that are presented in the manuscript are remarkably stable in their first and second order statistics, such as average spike rate, inter-spike intervals and pair-wise correlations.

Essential revisions:

The reviewers agreed that this manuscript provides a thorough and relatively complete description of a recording system and analytical framework that will likely set new benchmarks and therefore be of great interest to the neuroscience community. As set out in the following numbered points, the principal issues raised concerned the stability of the recordings and a need to compare the spike sorting algorithm, which combines established and innovative methods for spike clustering, with other approaches. The reviewers also found aspects of the spike sorting algorithm difficult to assess, and have requested more information about this. Finally, they were struck by how stable the recording parameters were over multiple days and wish to see how this compares for "earlier" trials (i.e. several thousand) versus really trials (i.e. after 10,000 to 20,000 trials).

1) The spike sorting method does not seem to be fully automatic, since there is a manual verification stage (merging of chains during "step F" appears to require manual input). The name adopted by the authors for this method (Fully Automated Spike Tracker) is therefore a little misleading and should probably change. More information is needed about the manual steps in order to avoid a lack of reliability when used by others. For example, there is no indication of how long the manual verification takes (e.g. in units of "human-hours per rat per day" or "number of manual chain merges/breaks required per tetrode per day"/"number of junk clusters manually excluded per rat per day"), which is central to the claim that the algorithm is an improvement on existing methods.

2) The authors say that manual merges of chains were done only in cases where ISI distributions had >0.9 correlation. How does this not invalidate Figure 5, since they seem to have selected units to be the "same" on the basis of this very property? Please explain this.

3) The field of spike sorting is plagued by a proliferation of papers that aren't compared against each other and often aren't even testable by the reader. Unfortunately this work is guilty on both counts: the results of the new algorithm are not compared to any existing algorithm (some of which do have drift-correction features, e.g. Calabrese and Paninski, 2011, J Neurosci Methods 196:159-169, and Jun et al., 2017, bioRxiv:1-29), and there is no way to test the code – following the github link, we get only an undocumented mess of folders with F# code; this needs to be fully explained. Independent of the paper's value scientifically, its value as a technical report is severely compromised by this oversight.

4) The initial clustering step of the algorithm (prior to any merging across time) is suspect since it uses a small number of spikes at a time (1000) and uses a clustering method (SPM) that, according to the authors, is typically hand-tuned for its key parameter (temperature). Here they use a multi-temperature method, but it ultimately comes down to a parameter "a" (subsection “Detailed algorithm for local clustering and de-noising”, point 3b) that enforces a similar trade-off between over- and under-clustering. The value of this parameter is not provided nor is there a description of the criteria on which its value was set. While the stability results probably could not have arisen if the spike sorting method failed to link related groups of spikes altogether, it certainly could have arisen if the groups of spikes (units) that were linked across time corresponded to over-clustered fractions of the true set of spikes from individual neurons, or if the units corresponded to under-clustered multi-unit activity (MUA). The argument that the units do not represent MUA rests on Figure 4, and specifically on the ISI quantification (since isolation distance can be high for a MU cluster that is well-separated from other clusters). But this quantification is difficult to interpret since the effective "censure period" of the spike detection algorithm is not quantified. This is actually not a minor detail: a spike with a width of, say, 0.75ms (not unusual for motor cortex) would be declared "ended" at the soonest 8 samples later, i.e. the next detected spike could not be detected <1ms after a broad waveform. For such units, the Fraction ISI <1ms will be zero *by construction*. The authors could quantify this easily: simulate a trace with gaussian noise and add the mean waveform of a neuron they recorded at completely random times. They could then run their spike detection directly on the trace and plot an ISI distribution of all detected spikes. If the mean waveform employed was broad, this will have <1% ISI <1ms, though in reality the data were created with no refractory period whatsoever. If that's the case, a longer refractory period should be used. At present, it appears that the authors have not demonstrated that their algorithm avoids over/under-clustering (a problem it seems likely to have), and an improved ISI analysis would help their case. Failing that, a discussion of the implications for the science if such errors were present would be helpful.

5) While the notion of accounting for apparently sparsely firing neurons is admirable, there are real concerns about whether these are indeed independent neurons. It seems equally likely that they could be spikes from an existing neuron that happens to stochastically fall outside of a cluster. Would defining clusters using >1000 spikes (e.g. 10,000) still result in this?

6) By clustering over 1000 spikes, and then over 1000 centroids, the algorithm puts a cap on the speed of probe-brain movement that can be tracked. Some discussion of this property would help readers to understand the algorithm, and even better would be if the authors were able to use their nice simulation from Figure 3 to quantify this: adjust the drift speed in the simulation and track how well the algorithm does for a range of speeds (this would also be useful for a range of firing rates).

7) The use of 50 microvolts for spike detection seems to be somewhat arbitrary. What is the typical noise level and the justification for this threshold? Was there an absolute upper-level threshold for a putative unit to eliminate artefacts?

8) The spike sorting method is a mixture of well-established algorithms and heuristics. While this is understandable, the heuristics have a number of "parameters" that clearly work well for the authors but might not generalize well to other systems. It would be useful to add a table of these parameters (tweaking knobs…) with range of values that you might recommend, so that readers and potential users are aware of these values. Here is the beginning of a long list: local chunks of 1000 spikes, 3x iteration for leftover spikes, 7 times the median of the absolute deviation for a threshold to get snippets, 15/1000 spikes for the leftover group, 1000 points for the second clustering, 10 cluster trees in the part 2, 0.02 threshold, etc.

9) How were the accelerometer data and the label of immobility (i.e. 2 SD threshold) validated?

10) The claims of stability are based on relatively simply behavioral markers. This is important because some of the cited papers used much more complex tasks in non-human primates. One might imagine more task-related variability for cue-dependent tasks with more degrees-of-freedom. Were the data analyzed during periods of very stable performance (e.g. after around 20,000 trials)? This might have a bearing on stability of neural activity, especially if the animals are in a crystallized and perhaps habitual mode. Did the authors analyze data from the earlier period of training (several thousand trials)? If so, was there any increased variability?

11) The discussion on the long-term stability of neural dynamics could be improved. The manuscript mentions that statistical frameworks could contribute (i.e. the Carmena 2005 and the Rokni papers). Also there is mention that methodological factors could contribute. It would be helpful to place their own findings in context and address these issues more directly. As mentioned above, could the over trained performance have contributed?

---

## [Author Response]

Essential revisions:The reviewers agreed that this manuscript provides a thorough and relatively complete description of a recording system and analytical framework that will likely set new benchmarks and therefore be of great interest to the neuroscience community. As set out in the following numbered points, the principal issues raised concerned the stability of the recordings and a need to compare the spike sorting algorithm, which combines established and innovative methods for spike clustering, with other approaches. The reviewers also found aspects of the spike sorting algorithm difficult to assess, and have requested more information about this. Finally, they were struck by how stable the recording parameters were over multiple days and wish to see how this compares for "earlier" trials (i.e. several thousand) versus really trials (i.e. after 10,000 to 20,000 trials).1) The spike sorting method does not seem to be fully automatic, since there is a manual verification stage (merging of chains during "step F" appears to require manual input). The name adopted by the authors for this method (Fully Automated Spike Tracker) is therefore a little misleading and should probably change. More information is needed about the manual steps in order to avoid a lack of reliability when used by others. For example, there is no indication of how long the manual verification takes (e.g. in units of "human-hours per rat per day" or "number of manual chain merges/breaks required per tetrode per day"/"number of junk clusters manually excluded per rat per day"), which is central to the claim that the algorithm is an improvement on existing methods.

Although FAST is largely automated, we agree with the reviewers that the need for a final manual verification step means that we cannot call it “Fully Automated Spike Tracker”. Our primary claim is that FAST greatly speeds up spike-sorting of continuously acquired neural data to make feasible truly long-term recordings from the same units over weeks and months. For these reasons, we have changed the name of our algorithm to “Fast Automated Spike Tracker”.

We apologize for the lack of information about the manual verification stage and now provide additional details in the Materials and methods section. It takes, on average, 8 human-hours to process 3 months of data recorded on a single tetrode, or ~130 hours for 16 tetrodes. This is about 5.5 times faster than the rate of data acquisition and is almost twice the rate of the slowest step (Step 1) in the automated processing pipeline. The number of manual mergers required per unit is, on average, 14.7 ± 30.6 (mean ± SD), which translates to an average of 1.3 manual chain mergers per tetrode per day. We had to manually split 2.0% of automatically clustered chains, corresponding to 0.027 chain splits per tetrode per day. This is now all documented in the manuscript.

Prompted by the reviewers’ comments, we have also illustrated our recommended workflow for the manual step in a new figure (Figure 2—figure supplement 3).

2) The authors say that manual merges of chains were done only in cases where ISI distributions had >0.9 correlation. How does this not invalidate Figure 5, since they seem to have selected units to be the "same" on the basis of this very property? Please explain this.

We thank the reviewers for highlighting this potential concern with the ISI stability analysis. We manually merged chains over relatively short time-gaps – less than 5 hours when the chains were part of the same continuous recording, or 24 hours if the chains originated from different recording files. Furthermore, when manually merging chains, we measured inter-chain similarity of ISI distributions using spikes recorded from the last hour of the preceding chain and the first hour of the succeeding chain. Thus, the process of manual merging does not prevent drift of the ISI distribution *within* an automatically defined cluster chain, or over time-scales longer than 24 hours.

To check if the ISI similarity requirement for chain merging biased our calculation of ISI stability, we performed the stability analysis at an earlier stage of the clustering process, on the chains generated by the fully automated steps 1-2 of FAST, i.e. prior to manual merging in step 3. The results are described in the text and shown in a new supplementary figure (Figure 5—figure supplement 1), and confirm our previous finding that ISI distributions are indeed stable over several days (Figure 5).

3) The field of spike sorting is plagued by a proliferation of papers that aren't compared against each other and often aren't even testable by the reader. Unfortunately this work is guilty on both counts: the results of the new algorithm are not compared to any existing algorithm (some of which do have drift-correction features, e.g. Calabrese and Paninski, 2011, J Neurosci Methods 196:159-169, and Jun et al., 2017, bioRxiv:1-29), and there is no way to test the code – following the github link, we get only an undocumented mess of folders with F# code; this needs to be fully explained. Independent of the paper's value scientifically, its value as a technical report is severely compromised by this oversight.

We fully agree with the reviewers that the performance of a novel spike-sorting algorithm should be tested against a known standard. For these reasons, we had applied FAST to both experimental and simulated ground-truth datasets, and compared its performance to the best ellipsoidal error rate (BEER), a metric that represents the best possible performance of standard spike-sorting algorithms that assume waveform stationarity.

The reviewers also recommended that we test the performance of FAST against other spike-sorting algorithms that have drift-correction features. FAST was specifically designed and optimized to process large, terabyte sized long-term datasets comprising upwards of 10^10^ spikes. Even our 10 day long simulated recordings that accurately model experimentally observed spike waveform drift alone comprise ~25 million spikes per tetrode. Although spike-sorting methods such as the Kalman filter mixture model (KFMM) approach (Calabrese and Paninski, 2011) offer interesting alternatives to FAST, they are not yet available as usable spike-sorting solutions. For instance, the KFMM release is proof-of-concept Matlab code that has not yet been optimized to deal with large numbers of spikes, and does not incorporate a method to automatically determine the appropriate number of clusters in a dataset. Under these circumstances, we feel that the extensive work required to get this algorithm to cluster our synthetic ground-truth data is beyond the scope of this study.

In contrast, the second spike-sorting algorithm cited by the reviewers (JRClust, Jun et al., 2017) has been optimized to run on large datasets. However, their approach to drift-correction is not a general solution to the problem of tracking non-stationary spike waveforms. Rather, their drift-correction algorithm is intimately tied to the design of their recording hardware – high-density ‘Neuropixels’ probes that allows them to spatially localize individual units along the surface of the electrode. The spatial localization afforded by such probes enables the use of a relatively simple drift correction algorithm for *global* motion of brain tissue relative to the probe. In contrast, FAST offers a more general *local* solution to drift correction that can work at the scale of single electrodes. Due to the additional hardware requirements for JRClust, we believe it is not directly comparable to spike waveform tracking by FAST.

We completely agree with the reviewers’ recommendation that we release FAST in a form that is easily testable by the reader. To that end, we have completely reorganized our GitHub repository (https://github.com/Olveczky-Lab/FAST) to provide a release of the FAST algorithm (v 1.0) that can even be installed on a standard desktop computer running the Windows OS. We have also written a detailed manual to provide installation and run time instructions. We successfully tested the clarity and completeness of these instructions by asking a naïve member of our lab to install FAST from scratch and then use it to parse a test dataset. Furthermore, to help readers visualize and perform manual merging and corrections on the automated output of FAST, we have also provided the code for a Matlab GUI we use in our manual verification step (https://github.com/Olveczky-Lab/FAST-ChainViewer). Finally, we would like to add that we are have recently hired a scientific programmer to develop FAST into an integrated, cross-platform program with an easy-to-use graphical user interface. We anticipate the release of FAST version 2.0 early next year.

4) The initial clustering step of the algorithm (prior to any merging across time) is suspect since it uses a small number of spikes at a time (1000) and uses a clustering method (SPM) that, according to the authors, is typically hand-tuned for its key parameter (temperature). Here they use a multi-temperature method, but it ultimately comes down to a parameter "a" (subsection “Detailed algorithm for local clustering and de-noising”, point 3b) that enforces a similar trade-off between over- and under-clustering. The value of this parameter is not provided nor is there a description of the criteria on which its value was set. While the stability results probably could not have arisen if the spike sorting method failed to link related groups of spikes altogether, it certainly could have arisen if the groups of spikes (units) that were linked across time corresponded to over-clustered fractions of the true set of spikes from individual neurons, or if the units corresponded to under-clustered multi-unit activity (MUA). The argument that the units do not represent MUA rests on Figure 4, and specifically on the ISI quantification (since isolation distance can be high for a MU cluster that is well-separated from other clusters). But this quantification is difficult to interpret since the effective "censure period" of the spike detection algorithm is not quantified. This is actually not a minor detail: a spike with a width of, say, 0.75ms (not unusual for motor cortex) would be declared "ended" at the soonest 8 samples later, i.e. the next detected spike could not be detected <1ms after a broad waveform. For such units, the Fraction ISI <1ms will be zero by construction. The authors could quantify this easily: simulate a trace with gaussian noise and add the mean waveform of a neuron they recorded at completely random times. They could then run their spike detection directly on the trace and plot an ISI distribution of all detected spikes. If the mean waveform employed was broad, this will have <1% ISI <1ms, though in reality the data were created with no refractory period whatsoever. If that's the case, a longer refractory period should be used. At present, it appears that the authors have not demonstrated that their algorithm avoids over/under-clustering (a problem it seems likely to have), and an improved ISI analysis would help their case. Failing that, a discussion of the implications for the science if such errors were present would be helpful.

We regret the misunderstanding that has arisen due to our inadequate descriptions of the rationale underlying various stages of the FAST algorithm. The initial clustering step in FAST can be more accurately thought of as a local ‘denoising’ procedure, whose outputs, the cluster centroids, represent a lossy compression of the raw spike dataset. We apologize for not providing more details about the parameter ‘a’ that is important for tuning the degree of this compression. As we now report in Table 1, the parameter ‘a’ was set to 20 µV, a value that was chosen empirically to favor over-clustering rather than under-clustering in order to prevent the incorporation of spikes from distinct units into the same local cluster. The rationale behind this stage is that Step 1 of FAST serves primarily to reduce the size and complexity of the dataset for the tracking algorithms implemented in Step 2 of FAST and does not, by itself, define the identity of the final single-unit clusters.

In Step 2, the cluster centroids computed in Step 1 are pooled into blocks of 1000 and then subjected to another round of superparamagnetic clustering across a range of different temperature values. This time, the cluster tree is not collapsed as in Step 1; instead we use a segmentation fusion algorithm to link together nodes of neighboring clustering trees that have high similarity, resulting in automatically defined single-unit cluster ‘chains’. Thus, Step 2 of FAST is actually the stage that defines cluster identity of centroid spikes and the relevant time-scale for single-unit clustering is defined by the span of the 1000 centroid block in Step 2 of FAST, not the 1000 spike block of the initial clustering step. In our datasets, Step 1 of FAST yields, on average, 8.0 ± 2.6 (mean ± SD, n = 80 tetrodes) cluster centroids per 1000 spike block. This means that a 1000 centroid block effectively encompasses ~125 blocks in Step 1, corresponding to 125000 spikes. Therefore, our multi-scale clustering approach ensures that single unit clusters are actually defined over relatively large windows spanning >100000 spikes, thereby resulting in accurate clustering.

We have modified our descriptions of the FAST algorithm in the Results and Methods sections to more effectively convey these points.

We thank the reviewers for raising an important concern about our spike-sorting quality check. Prompted by their comments, we estimated the censure time for our spike-detection method from the mean waveform of all isolated units, calculated in hour long time-bins. Indeed, we found that the censure time can be rather long – on average, 0.74 ± 0.07 ms and 0.46 ± 0.1 ms for medium spiny neurons and fast-spiking interneurons in the striatum, and 0.68 ± 0.07 ms and 0.44 ± 0.08 ms for regular spiking and fast spiking units in the motor cortex (mean ± SD), respectively. As the reviewers correctly point out, this would artificially reduce the fraction of ISIs < 1 ms, especially for regular spiking and MSN units.

As suggested, we revised our measures of unit quality using a longer 2 ms definition of the absolute refractory period (Figure 4). This caused a minor reduction in the fraction of hourly unit-time bins that passed the quality check (5.4% fewer bins), so we redid all our population-level analyses of long-term stability of unit activity (presented in Figure 5–Figure 8 and their supplemental figures). Our earlier finding that firing rates, inter-spike interval histograms, correlations and behavioral representations are stable over 10-20 day time-scales remains unchanged.

5) While the notion of accounting for apparently sparsely firing neurons is admirable, there are real concerns about whether these are indeed independent neurons. It seems equally likely that they could be spikes from an existing neuron that happens to stochastically fall outside of a cluster. Would defining clusters using >1000 spikes (e.g. 10,000) still result in this?

We thank the reviewers for bringing up this important point. We were primarily concerned about two kinds of outliers – ‘stochastic’ outliers that fall outside cluster boundaries by chance due to background noise, and ‘systematic’ outliers that arise due to changes in spike amplitude during bursts or coincident spiking of two or more units with distinct waveforms. Regardless of their origin, we can easily detect an outlier cluster in the manual verification step (described in Materials and methods and Figure 2—figure supplement 3 of the revised manuscript) by the presence of a refractory period in spike-timecross-correlograms and close resemblance of spike waveforms with respect to a ‘parent’ cluster.

We rarely observe chains of stochastic outliers in our datasets. This is due to two factors. First, we impose a minimum cluster size of 15 spikes for every 1000 spike block in FAST’s local clustering step (Step 1). Per definition, stochastic outlier spikes are unlikely to have similar waveforms to each other and are therefore unlikely to form local clusters larger than this 15 spike threshold. Second, the cluster centroids obtained in Step 1 are subject to another round of clustering in Step 2 so as to merge multiple centroids that correspond to the same unit, as described in our response to reviewer comment #4. The factors that are more relevant in determining the extent of stochastic outliers are (1) the parameter ‘a’, which tunes the degree of under/over-clustering of a unit’s spikes into different local clusters, (2) the minimum cluster size threshold, and (3) the effective number of spikes encompassed by the 1000 centroid block in Step 2 of FAST.

Occasionally, we do observe short ‘satellite’ chains comprising stochastic outliers of parent units that have high firing rates. In the manual verification step, we merge any outlier chains we have identified with the parent cluster. Systematic outliers are more commonly observed in our datasets. We manually merge pairs of units with identical spike shapes but distinct spike amplitudes, if their spike-time cross-correlograms are highly asymmetric around 0 ms and show a distinct refractory period, as these distinct clusters most likely arose from the same bursting unit. Finally, we reject all clusters with complex waveforms that are the result of simultaneous spiking of two or more units, as it is a computational challenge to identify the parent units. In the future, we hope to extend FAST to deal with simultaneous spiking of multiple units by incorporating ideas from recent efforts to tackle this problem (Ekanadham et al., 2014).

We have added a description of our procedures to deal with outlier spikes to the Materials and methods section of the revised manuscript.

6) By clustering over 1000 spikes, and then over 1000 centroids, the algorithm puts a cap on the speed of probe-brain movement that can be tracked. Some discussion of this property would help readers to understand the algorithm, and even better would be if the authors were able to use their nice simulation from Figure 3 to quantify this: adjust the drift speed in the simulation and track how well the algorithm does for a range of speeds (this would also be useful for a range of firing rates).

This is a great suggestion. We have added a paragraph on estimating the time-scales of drift correction by FAST to the Materials and methods section of our manuscript:

“The time-period over which FAST assumes that spike waveforms are stable (the ‘tracking time-scale’) is a function of both the mean event (spike) detection rate and the number of distinct units simultaneously recorded on the electrode. […] This is because FAST employs superparamagnetic clustering which can also sort the non-Gaussian shaped clusters that are an outcome of electrode drift (Quiroga et al., 2004; Rey et al., 2015).”

7) The use of 50 microvolts for spike detection seems to be somewhat arbitrary. What is the typical noise level and the justification for this threshold? Was there an absolute upper-level threshold for a putative unit to eliminate artefacts?

The average noise level (median absolute deviation*1.4826) in our recordings was 11.3 ± 0.9 µV (mean ± SD), with very little variation across recording electrodes. Our 50 µV spike detection threshold is approximately 5 times this noise level. We note that this value is not hard-coded in FAST and can be easily altered to a multiple of the measured noise; we simply used this number for convenience. We have updated the text to include this information.

We did not have an absolute upper-level threshold to eliminate artefacts as we sometimes recorded spikes with large amplitudes exceeding 2 mV. Instead, artefacts that did not resemble spikes were eliminated in the subsequent clustering steps of the FAST algorithm.

8) The spike sorting method is a mixture of well-established algorithms and heuristics. While this is understandable, the heuristics have a number of "parameters" that clearly work well for the authors but might not generalize well to other systems. It would be useful to add a table of these parameters (tweaking knobs…) with range of values that you might recommend, so that readers and potential users are aware of these values. Here is the beginning of a long list: local chunks of 1000 spikes, 3x iteration for leftover spikes, 7 times the median of the absolute deviation for a threshold to get snippets, 15/1000 spikes for the leftover group, 1000 points for the second clustering, 10 cluster trees in the part 2, 0.02 threshold, etc.

We thank the reviewers for this excellent suggestion. We have prepared a new table (Table 1) listing the FAST parameter values that we found to work well for both the dorsolateral striatum and motor cortex.

9) How were the accelerometer data and the label of immobility (i.e. 2 SD threshold) validated?

We used video data recorded by the overhead web camera to validate this. Active and inactive periods of behavior were scored from >8 hours of video (n=2 rats) at 1 s resolution by a human observer. The agreement between automatic and manual classifications was 94.3% . We have now included a table in the manuscript, showing the overlap between inactive and active periods in the manual and automatic classifications.

10) The claims of stability are based on relatively simply behavioral markers. This is important because some of the cited papers used much more complex tasks in non-human primates. One might imagine more task-related variability for cue-dependent tasks with more degrees-of-freedom. Were the data analyzed during periods of very stable performance (e.g. after around 20,000 trials)? This might have a bearing on stability of neural activity, especially if the animals are in a crystallized and perhaps habitual mode. Did the authors analyze data from the earlier period of training (several thousand trials)? If so, was there any increased variability?

We thank the reviewers for the opportunity to clarify our views on this topic.

We agree that the behaviors we train our rats on are, in some aspects, less complex than the tasks performed by non-human primates (NHPs) in previous studies, since the latter behaviors tend to be cue-driven and require greater trial-to-trial flexibility in motor planning. However, in other ways, our behavioral paradigms can be considered to be more complex in that animals generate sequences of precisely timed actions in contrast to the single ballistic movements performed by NHPs in center-out reaching tasks. In addition to the prescribed task (Figure 8), our study also assays motor representations of many naturalistic behaviors such as grooming, active exploration and eating (Figure 8). Furthermore, most studies in non-human primates have usually reported on stability of low-dimensional features of motor representations such as spike-count tuning curves or preferred movement direction. In contrast, we assay the stability of higher dimensional measures of motor representations such as spike-triggered averages and peri-event time histograms. In light of ongoing debates about motor coding in areas such as the motor cortex (Churchland et al., 2012; Georgopoulos et al., 1986; Kakei et al., 1999; Todorov, 2000) and striatum (Desmurget and Turner, 2010; Dudman and Krakauer, 2016; Graybiel, 1998), we believe that our measures make fewer assumptions about the encoding schemes in these circuits.

The animals used in our study had achieved expert performance on the interval pressing task prior to implantation of electrode arrays. The rationale behind this decision was that we wanted to assay stability of motor representations under conditions of stable behavior. We agree that it would be interesting to assay the stability of neural activity during the early learning phase of the task and we are actively pursuing this question in our laboratory, but we believe this experiment goes beyond the scope of the present study for the following reasons. First, many prior studies have reported instability of single neuron representations even under conditions of stable behavior (Carmena et al., 2005; Huber et al., 2012; Libertine et al., 2016; Mankin et al., 2012; Morcos and Harvey, 2016; Rokni et al., 2007; Ziv et al., 2013). Second, rats’ movement sequences in the early phase of the interval-pressing task are highly variable from one trial to the next (Kawai et al., 2015). Without knowing the neural code for movement in the motor cortex and striatum, this increased behavioral variability can confound measurements of the stability of motor representations.

11) The discussion on the long-term stability of neural dynamics could be improved. The manuscript mentions that statistical frameworks could contribute (i.e. the Carmena 2005 and the Rokni papers). Also there is mention that methodological factors could contribute. It would be helpful to place their own findings in context and address these issues more directly. As mentioned above, could the over trained performance have contributed?

We thank the reviewers for this suggestion. We have included the following paragraph to our discussion of long-term stability of neural dynamics.

“Discrepancies between our findings and other studies that report long-term instability of neural activity could, in some cases, be due to inherent differences between brain areas. […] Our experimental platform makes it feasible to characterize learning-related changes in task representations and relate them to reorganization of network dynamics over long time-scales.”